# Sub-token ViT Embedding via Stochastic Resonance Transformers

## Abstract

We discover the presence of quantization artifacts in pre-trained Vision Transformers (ViTs), which arise due to the image tokenization step inherent in these architectures. These artifacts result in coarsely quantized features, which negatively impact performance, especially on downstream dense prediction tasks. We present a zero-shot method to improve how pre-trained ViTs handle spatial quantization. In particular, we propose to ensemble the features obtained from perturbing input images via sub-token spatial translations, inspired by Stochastic Resonance, a method traditionally applied to climate dynamics and signal processing. We term our method "Stochastic Resonance Transformer" (SRT), which we show can effectively super-resolve features of pre-trained ViTs, capturing more of the local fine-grained structures that might otherwise be neglected as a result of tokenization. SRT can be applied at any layer, on any task, and does not require any fine-tuning. The advantage of the former is evident when applied to monocular depth prediction, where we show that ensembling model outputs are detrimental while applying SRT on intermediate ViT features outperforms the baseline models by an average of $4.7\%$ and $14.9\%$ on the RMSE and RMSE_log metrics across three different architectures. When applied to semi-supervised video object segmentation, SRT also improves over the baseline models uniformly across all metrics, and by an average of $2.4\%$ in F&J score. We further show that these quantization artifacts can be attenuated to some extent via self-distillation. On the unsupervised salient region segmentation, SRT improves upon the base model by an average of 1.8% on the maxF metric. Finally, we show that despite operating purely on pixel-level features, SRT generalizes to non-dense prediction tasks such as image retrieval and object discovery, yielding consistent improvements of up to $2.6\%$ and $1.0\%$ respectively.

## 1 Introduction

The Transformer architecture, that takes quantized or "tokenized" inputs, seems ill fit for vision tasks, since images do not have a natural discretization scale: The same object can disappear within a pixel or fill the entire image plane depending on its distance from the camera. Yet Vision Transformers (ViTs) have been shown effective especially in semantic inference tasks, so we focus on simple methods to use pre-trained ViT models while addressing some of the shortcomings of a fixed spatial quantization of the input tokens.

The standard remedy for quantization artifacts is anti-aliasing. In one-dimensional signals such as audio, anti-aliasing refers to averaging nearby samples, or equivalently translated versions of the signal. For images, in addition to quantization of the translation group reflected in the size of the pixels, there is also the scale (semi-)group, reflected in the size of projection of objects onto the image plane. Various network architectures comprise spatial average pooling, which is translational anti-aliasing, and the notion of domain-size pooling has been introduced in local features by (Dong & Soatto, 2015). While traditionally antialiasing is performed via convolution with a fixed kernel, Stochastic Resonance simply perturbs the data with respect to an artificial distribution and then averages the results. Stochastic resonance can be thought of as a way of performing data augmentation, or adaptive quantization. This simplistic approach is well suited to pre-trained transformers since it only requires acting on inputs and outputs without modifying (or even knowing) the weights of the model. Stochastic Resonance is used to resolve coarsely quantized signals beyond the native

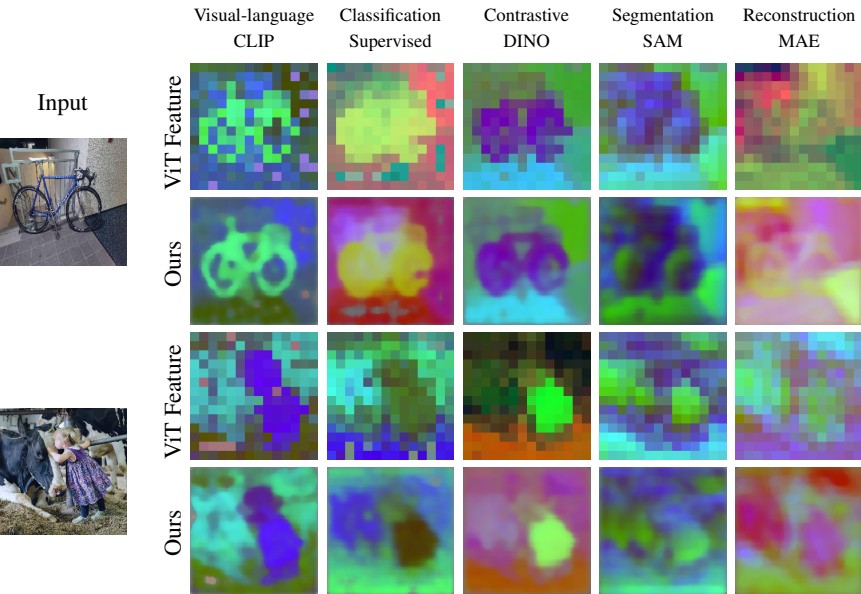

Figure 1: **High-resolution ViT features Computed by Stochastic Resonance.** *Stochastic Resonance enables super-resolving tokenized ViT features during inference without the need for additional training or modifying ViT forward pass. Features visualized via Principal Component Analysis are helpful in analyzing different pre-trained ViT models: CLIP (Radford et al., 2021) captures major image components. Interestingly, although Supervised (Dosovitskiy et al., 2020) and DINO (Caron et al., 2021) are trained by different pipelines and training loss, they prioritize similar regions. This may be due to they are trained on the same dataset and thus capture similar inductive bias. In contrast, SAM (Kirillov et al., 2023) and MAE (He et al., 2022) capture local features over high-level semantics. Stochastic Resonance not only serves as a powerful visualization tool but also enhances model performance across multiple downstream tasks, as demonstrated in Sect. 3.*

resolution of the sensor and has found wide applicability in cochlear implants. We apply the same mechanism to the spatial dimension, by perturbing the input signal to a ViT, which results in highly variable outcomes for the embeddings. Such outcomes are aggregated statistically to first-order (mean or median) and second-order (dispersion) to yield a sub-token embedding along with a measure of confidence or consistency, which have broad potential applications. For instance, the median outcome can be used as a seed for unsupervised object segmentation (along with motion masks), and the dispersion can be used as a weight for an adaptive regularizer.

We call the resulting methods "Stochastic Resonance Transformer" although we do not modify the transformer itself. Instead, we can leverage ViTs, pre-trained on large datasets, such as CLIP and DINO, to improve their handling of spatial quantization. This may help attenuate some of the biases of these datasets, for instance the object-centric nature of DINO, which biases the representation towards centered objects that occupy a large portion of the visual field. Stochastic Resonance Transformer can be used to super-resolve and ensemble the feature maps in ViT, outputting features that reveal some of the local fine-grained image structure. This can be done at any ViT layer, on any task, without altering network architecture or pre-trained network weights. We can optionally distill the fine-grained features back to the original ViT scale, where we notice performance increase at equal inference time and cost. Our contributions are listed as follows:

- We introduce a novel technique, namely the Stochastic Resonance Transformer (SRT), that super-resolves ViT embeddings without additional training or modifications to the ViT's forward pass.
- SRT yields a versatile visualization tool that can be applied to any layer of any pre-trained ViT model, offering valuable insights into ViT model characteristics.
- The enhanced embeddings from SRT can be seamlessly integrated into any task that utilizes ViT as a feature extractor, thereby serving as a test-time augmentation and ensemble method.
- We showcase the effectiveness of SRT by consistent improvement on a range of diverse vision tasks. Notably, it demonstrates significant enhancements on dense prediction tasks, of up to 14.9% on depth prediction, 2.4% on video object segmentation, and 1.8% on salient region segmentation.
- We provide an efficient implementation SRT, including parallelization and recursive aggregation, which reduces computational and memory requirements.

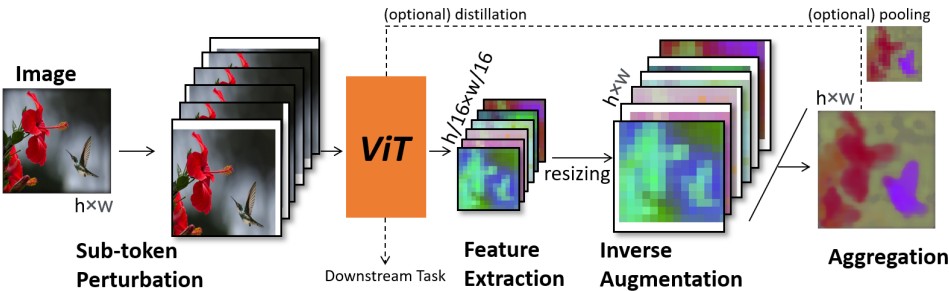

Figure 2: **Schematic for SRT.** *SRT applies controlled perturbations via translations to input images, extracting features through Vision Transformers (ViTs). These features are then upsampled to higher resolution and aligned using the inverse of the applied perturbations. Statistical aggregation, including mean and median, along the perturbation dimension, produces fine-grained feature representations. These features find utility in visualization and can also be seamlessly integrated back into the network for enhanced performance in downstream tasks.*

## 2 METHOD

In this section, we first describe the conceptual framework and pipeline of SRT in Sect. 2.1, then we formalize SRT in Sect. 2.2, and present details for efficient implementation in Secton 2.3

### 2.1 SUPER-RESOLVING VIT EMBEDDINGS BY STOCHASTIC RESONANCE.

Given an image $x$ with $N \times M$ resolution, a Vision Transformer (ViT) divides it into tokens, where each token represents a $n \times m$ rectangular patch. While tokens can technically overlap, practical ViT models often use non-overlapping tokens for efficiency due to the quadratic complexity of transformers with respect to the number of tokens. Consequently, in a certain layer of ViT, this approach yields a feature map with dimensions $\frac{N}{n} \times \frac{M}{m} \times C$, where $C$ is the size of the feature vector determined by architecture, downsampled from the original image and subsequently losing sub-token spatial information.

Given a trained ViT model, we aim to obtain features in a higher resolution that preserves the spatial information on a pixel level, ideally matching with the original image input. Fig. 2 illustrates our proposed pipeline of SRT. To achieve super-resolution of the features, we introduce random sub-token perturbation to the input, i.e. transforming the coordinates of the input and resampling onto a new image plane, and extract embeddings from the resulting perturbed image. We then upsample the resulting low-resolution embeddings back to the original image resolution $N \times M$ and apply an inverse of the perturbation to the spatial coordinates of the embeddings, and through an inverse warp, align it with the original input image.

By repeating this process on different sub-token perturbations for $t$ times, we generate a collection of embeddings, denoted by $N \times M \times C \times t$, that are spatially aligned to the input frame of reference. We can then compute statistics, e.g. mean or median, along the $t$ dimension. Consequently, we obtain a feature field $N \times M \times C$, with the same spatial resolution as the original image. As showcased in Fig. 6, the embeddings are "super-resolved" to sub-token resolution. This process is similar to Stochastic Resonance, where introducing white noise to the input signal super-resolves a signal beyond the native resolution. These embeddings offer promising downstream applications, as discussed in more detail in Sect. 3.

For any task that utilizes ViT as a feature extractor, we can take an additional step by applying average pooling to again tokenize this high-resolution feature, to map it to $\frac{N}{n} \times \frac{M}{m} \times C$. It's important to note that this feature differs from the one obtained from one single forward pass of ViT, as it is an aggregate of multiple perturbed inputs. This process can be viewed as test-time augmentation and ensemble. Since this feature is compatible with the original ViT architecture, it can be seamlessly integrated into any model at any layer, regardless of pre-training, without requiring additional learned modules or altering the forward pass. Such a pipeline improves performance on diverse computer vision tasks, as validated by Sect. 3. Next, we formalize the aforementioned pipeline.

## 2.2 FORMALIZATION

$x \in \mathbb{R}^{N \times M \times K}$ is a $K$-channel signal (*e.g.*, $K = 3$ for a color image.) Let $\pi : \mathbb{R}^{N \times M} \to \mathbb{R}^{n \times m}; x \mapsto x$ a projection (subsampling, $n \ll N, m \ll M$), with the corresponding inverse (interpolation) map $\pi^{-1} : \mathbb{R}^{n \times m} \to \mathbb{R}^{N \times M}; x \mapsto x$ be piecewise constant. This is a trivial form of subsampling and interpolation with a constant kernel.

Now, let $\phi : \mathbb{R}^{NMK} \to \mathbb{R}^{nmC}$ a trained model with $C$ channels of feature maps, typically $C \gg K$. Finally, let $T : \mathbb{R}^{N \times M} \to \mathbb{R}^{N \times M}; x \mapsto Tx$ a compact and invertible transformation, for instance, e padded shift by a number of pixels smaller than $(N - n)/n \times (M - m)/m$. We consider uniform random padded shifts (translation) and consider the following measurement process:

$$y_t = \phi(T_t x) \tag{1}$$

for all random transformations $T_t$. We wish to super-resolve[1] the output of $\phi$ from $n \times m$ to $N \times M$. We do so iteratively by averaging (or by a trainable linear transformation $K_t$) with respect to the innovation process:

$$\epsilon_t = \underbrace{\pi \left( T_t^{-1} \pi^{-1} y_t \right)}_{\hat{y}_t} - K_t \phi(x) \tag{2}$$

now the super-resolved features which we call $\hat{x}_t$ are obtained by an observer architecture, which implements a closed-loop dynamical system of the form:

$$\begin{cases} \hat{x}_{t+1} = \hat{x}_t + T_t^{-1} \pi^{-1} y_t & \hat{x}_0 = 0; \\ y_t = \phi(T_t x) \end{cases} \tag{3}$$

This is just a super-resolved moving average, whereby the variance of $\hat{x}$ will decrease to a steady state (by Central Limit Theorem), following the practice of stochastic resonance. It is a mixture of upsampling/interpolation and inference-time data augmentation, or ensembling.

## 2.3 EFFICIENT IMPLEMENTATION

In theory, there is no limitation on the types of sub-token transformations that can be employed. We opt for a straightforward approach by applying translations (with padding) and this practice demonstrates effective results. We sample translations at the pixel level, avoiding the need for sub-pixel interpolation, which could introduce unwanted artifacts.

For a ViT utilizing token sizes of $m \times n$, we impose a constraint on the maximum magnitude of translation, limiting it to $\frac{m}{2} \times \frac{n}{2}$. This constraint allows the model to explore all possible token selections within the image. It is worth noting that excessive translation can be counterproductive when applied to downstream vision tasks, as it can result in information loss at the image boundaries. A detailed discussion can be found in Sect3.2, where we study the relation between perturbation level and model performance.

Furthermore, our framework facilitates computational acceleration through batching. Each ViT forward pass on different augmented images operates independently, enabling parallelization. Nevertheless, in practical vision tasks, GPU memory constraints may pose challenges. This is particularly true after upsampling the embeddings, as each $M \times N \times C$ tensor consumes a significant amount of memory, and there could be numerous such tensors. To address this issue, we employ recursive mean computation to iteratively aggregate information. In the cases where the sole objective is to obtain the embeddings after average pooling, we can simplify the pipeline by even bypassing the upsampling step. Instead, we explicitly hard-code the average pooling of SRT in a recursive manner, by exploiting the property of translation as one averaged token can be explicitly computed by 4 neighboring tokens[2]. This approach enhances inference speed by a factor of 25 compared to a naive implementation of SRT and substantially reduces GPU memory usage. With ViT-16/S architecture, on DAVIS-2017 (Pont-Tuset et al., 2017) our implementation of SRT runs at 2.2 seconds per image on a Nvidia 1080Ti GPU using a perturbation level of 3 pixels, which is 14 times slower than running a single forward pass, despite SRT requires 49 independent ViT forward passes. To further speed up, one may optionally fine-tune the ViT model by distilling utilizing SRT, so that the inference time and cost remain, as demonstrated in Sect. 3.2.

---

[1]We call this process *immersion* since each point $x$ maps to $z = \phi(x)$ but $z \neq T^{-1}\phi(Tx)$. In other words, $x$ is mapped injectively but not bijectively, since there are as many (vector)-values as sampled value of $T$.

[2]Our implementation will be publicly available upon paper publication.

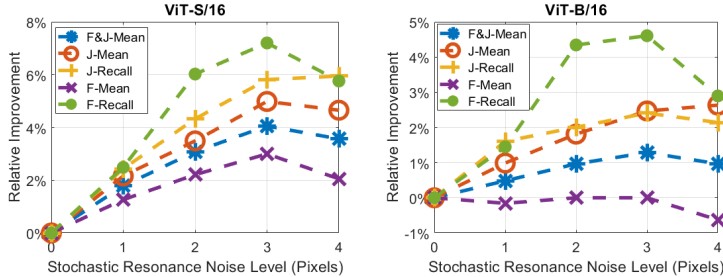

Figure 3: **Relative improvement on DAVIS-2017 dataset vs different noise levels.** *There exists an inherent trade-off between perturbation level and performance gain. Smaller perturbation ranges result in weaker improvements from the baseline model due to lower input diversity, while larger perturbations are susceptible to greater information loss. 3 pixels is found to be the optimal perturbation level on both ViT-S/16 and Vit-B/16.*

## 3 EXPERIMENTS

We first showcase SRT as a powerful visualization tool. We use it to visualize fine-grained embeddings from well-established ViT models, illustrating SRT's capacity to provide insights into the properties of ViTs. We then apply SRT to various downstream tasks that utilize ViT as a backbone, employing SRT in a zero-shot manner without fine-tuning. This evaluation validates SRT's ability to improve diverse vision tasks, and the ensembled embeddings seamlessly integrate with the original pipeline. Remarkably, SRT consistently improves performance across all five tasks.

### 3.1 DENSE ViT FEATURE VISUALIZATION

SRT demonstrates significant promise in visualizing features of ViT models. It achieves this without necessitating modifications to the ViT's forward pass. In Fig. 6, we present visualizations of the final layer features from five popular ViT models, all employing the ViT-B/16 architecture. Notably, all visualizations are computed by a standard consumer laptop. We employ SRT with a turbulence level of 7 pixels to traverse non-overlapping augmented tokens extensively. The resultant high-dimensional features then go through Principal Component Analysis (PCA), with the top three components mapped to RGB channels to facilitate effective visualization. Despite sharing the same architecture, the five models exhibit distinct characteristics owing to variations in their pre-training supervision. For instance, CLIP (Radford et al., 2021) is trained through contrastive visual-language pre-training and captures major image components in the displayed examples. The Supervised model (Dosovitskiy et al., 2020) is trained for ImageNet classification, while DINO (Caron et al., 2021) undergoes contrastive learning. Interestingly, despite their diverse training regimes, both models prioritize similar image regions, potentially due to their shared dataset and resulting common inductive bias. In contrast, SAM (Kirillov et al., 2023) is trained on massive segmentation masks without semantic labels or object-centric priors, and MAE (He et al., 2022) is trained through inpainting of randomly masked image regions. Both methods emphasize local image features over high-level semantics. Our versatile visualization tool provides valuable insights into the characteristics of ViT models, offering substantial potential for practical applications.

### 3.2 SEMI-SUPERVISED VIDEO OBJECT SEGMENTATION

We apply SRT to evaluate its performance using the DAVIS-2017 video instance segmentation benchmark (Pont-Tuset et al., 2017). We adhere to the experimental methodology established in (Jabri et al., 2020), which employs a "semi-supervised" video object segmentation approach on the original 480p resolution. Provided with the initial annotation of the objects of interest in the first frame, this method subsequently propagates the segmentation between consecutive frames. Notably, the method utilizes the last layer feature of the Vision Transformer (ViT) to guide this segmentation propagation process. Consequently, the quality of the ViT features directly impacts the final segmentation results. For optimal outcomes, these features must possess discriminative and semantically meaningful characteristics to effectively support this segmentation task.

| Method | F&J-Mean | J-Mean | J-Recall | F-Mean | F-Recall |
|---|---|---|---|---|---|
| DINO-ViT-S/16 | 0.617 | 0.602 | 0.740 | 0.634 | 0.764 |
| + SRT | **0.642** | **0.632** | **0.783** | **0.653** | **0.819** |
| Distill by SRT | 0.625 | 0.609 | 0.745 | 0.642 | 0.780 |
| + Overlapping tokens | 0.591 | 0.577 | 0.706 | 0.605 | 0.741 |
| + Naive ensemble | 0.477 | 0.455 | 0.468 | 0.500 | 0.542 |
| DINO-ViT-B/16 | 0.622 | 0.608 | 0.748 | 0.637 | 0.760 |
| + SRT | **0.630** | **0.623** | **0.766** | **0.637** | **0.795** |
| DINO-ViT-S/8 | 0.706 | 0.675 | 0.815 | 0.737 | 0.846 |
| + SRT | **0.720** | **0.688** | **0.827** | **0.752** | **0.868** |

Table 1: **Results on DAVIS-2017 video object segmentation.** *Applying SRT improves over the baseline models uniformly over all metrics, as measured across 3 variants of ViTs trained using the DINO (Caron et al., 2021) contrastive learning objective. SRT yields significant improvements even for ViT-S/8 trained with finer patch sizes (8x8). One may optionally fine-tune the original ViT model by distilling by SRT, which increases performance while inference time and cost remain one single forward pass.*

In our study, we evaluate various Vision Transformer (ViT) models pre-trained using the DINO (Caron et al., 2021) contrastive scheme. We adopt three different architectures, specifically ViT-S/16, ViT-B/16, and ViT-S/8, each varying in their spatial patch size (16x16 pixels and 8x8 pixels). Our results in Tab. 1 indicate that, on average, SRT enhances the original baseline models by a relative 2.4% in terms of the F&J score. The most significant improvement is observed with ViT-S/16, where the improvement reaches 4.1%. Importantly, these enhancements are achieved without any modifications to the model or pre-trained weights. However, we address a potential criticism of our approach, which could be seen as trivial test-time augmentation combined with feature-level ensemble. To counter this concern, we perform a heuristic by naively augmenting images by color jitter and performing feature-level ensemble, and we find that this method is, in fact, detrimental to performance. We also reproduce the approach proposed by Amir et al. (2021) that uses overlapping tokens at inference time, which negatively impacts the results. We investigate whether inference costs induced by SRT can potentially be mitigated via distillation. Toward this goal, we attempt to learn the ensembled SRT representations using the following self-distillation objective:

$$\min_w \sum_{x \in \mathcal{D}} ||\phi_w(x) - SRT(x, w_0)||, \tag{4}$$

where $\phi$ and $(w_0)$ $w$ are the ViT and its (original) parameters, and $x$ the image in the target dataset. Our preliminary results on DINO-ViT/16 improve from the baseline by $1.3\%$ after the self-distillation step. Note that Eq. equation 4 is agnostic to the task and requires no label, rendering distillation by SRT a fine-tuning scheme that adapts pre-trained ViT features to new target datasets. We leave the investigation of this to future work.

Fig. 3 illustrates the relative improvement across different perturbation levels of SRT applied to ViT-S/16 and ViT-B/16. While higher perturbation levels offer greater input diversity, they are also susceptible to information loss. We anticipate a trade-off between perturbation level and performance gain and empirically identify a perturbation level of 3 pixels as the optimal point for both.

### 3.3 MONOCULAR DEPTH PREDICTION

We extend the application of SRT to monocular depth estimation, a task that leverages ViT features from multiple ViT layers, in contrast to video object segmentation which primarily utilizes the last layer features. This choice of task highlights the versatility of SRT, showcasing its seamless compatibility with various ViT layers and architectures. Specifically, we evaluate three ViT architectures: ViT-S/14, ViT-B/14, and ViT-L/14, each equipped with two prediction heads (linear and DPT (Ranftl et al., 2021)). We adopt the experimental settings provided by DINOV2, which offers pre-trained backbones and corresponding prediction heads. Our assessment utilizes the NYU-V2 dataset (Nathan Silberman & Fergus, 2012) under its original $640 \times 480$ resolution.

Tab. 2 presents the results, demonstrating consistent improvements over baseline methods. The most significant enhancements are observed in the RMSE and RMSE_log metrics, where we achieve relative improvements of 4.7% and 14.9% with linear heads, and 3.6% and 11.0% with DPT heads,

| Backbone | Head | Method | RMSE | RMSE_log | AbsRel | SqRel | a1 | a2 | a3 |
|---|---|---|---|---|---|---|---|---|---|
| DINOV2-ViT-B/14 | Linear | Baseline | 0.396 | 0.135 | 0.100 | 0.061 | 0.903 | 0.983 | 0.996 |
| | | +OE | 0.376 | 0.121 | 0.093 | 0.059 | 0.918 | 0.984 | 0.997 |
| | | +SRT | **0.349** | **0.108** | **0.087** | **0.052** | **0.930** | **0.990** | **0.998** |
| | DPT | Baseline | 0.323 | 0.109 | 0.074 | 0.044 | 0.941 | 0.987 | 0.996 |
| | | +OE | 0.314 | 0.101 | **0.073** | **0.043** | 0.944 | 0.988 | **0.997** |
| | | +SRT | **0.305** | **0.096** | **0.073** | **0.043** | **0.945** | **0.989** | **0.997** |
| DINOV2-ViT-S/14 | Linear | Baseline | 0.471 | 0.162 | 0.125 | **0.084** | 0.853 | 0.972 | 0.994 |
| | | +OE | 0.486 | 0.153 | 0.126 | 0.095 | 0.858 | 0.974 | 0.994 |
| | | +SRT | **0.457** | **0.140** | **0.118** | 0.085 | **0.876** | **0.980** | **0.996** |
| | DPT | Baseline | 0.336 | 0.114 | 0.080 | **0.048** | 0.933 | 0.986 | 0.996 |
| | | +OE | 0.347 | 0.114 | 0.080 | 0.053 | 0.932 | 0.985 | 0.996 |
| | | +SRT | **0.334** | **0.104** | 0.080 | 0.051 | **0.935** | **0.988** | 0.996 |
| DINOV2-ViT-L/14 | Linear | Baseline | 0.373 | 0.127 | 0.093 | 0.054 | 0.916 | 0.985 | 0.996 |
| | | +OE | 0.401 | 0.131 | 0.097 | 0.062 | 0.908 | 0.982 | 0.996 |
| | | +SRT | **0.365** | **0.113** | **0.090** | **0.053** | **0.924** | **0.989** | **0.998** |
| | DPT | Baseline | 0.311 | 0.105 | **0.070** | 0.042 | 0.946 | 0.988 | **0.997** |
| | | +OE | 0.317 | 0.103 | 0.072 | 0.044 | 0.942 | 0.987 | 0.996 |
| | | +SRT | **0.297** | **0.092** | **0.070** | **0.041** | **0.947** | **0.991** | **0.997** |

Table 2: **Results on NYU-V2 depth prediction.** *Our method can be extended without modification to improve intermediate features to yield improved performance on the downstream depth prediction tasks. While ensembling of outputs (OE) can often be detrimental to performance, applying SRT on the features from pretrained backbones (inputs to prediction heads) can improve performance over baselines by $4.7\%$ and $14.9\%$ on RMSE and RMSE_log, using the linear prediction head and by $3.6\%$ and $11.0\%$ using the DPT head.*

respectively. Notably, these metrics are sensitive to outliers, highlighting the effectiveness of our approach in mitigating instability in ViT features and enhancing robustness. For completeness, we compare our method with output-space ensemble (marked as "OE"), which employs the perturbations as SRT, and aggregates the model output instead of intermediate features. We find no significant improvements, and in some cases, this method is even detrimental. This underscores the robustness of SRT's feature ensemble scheme.

### 3.4 Unsupervised Salient Region Segmentation

We employ SRT in conjunction with TokenCut (Wang et al., 2022) for unsupervised salient region segmentation tasks. TokenCut is a graph-based approach that applies the Normalized Cut algorithm to partition ViT tokens into two distinct clusters, representing the salient foreground and the background. The key challenge is to ensure that the features are not only discriminative across clusters but also consistent within clusters.The results are in Tab. 3. We adopt three datasets: ECSSD (Shi et al., 2015), DUTS (Wang et al., 2017), and DUT-OMRON (Yang et al., 2013), following the TokenCut, and consistently observe improvements, with an average increase in the maxF metric of 1.8%. Notably, this improvement is constrained by the model architecture, as it operates at the coarse segmentation level of ViT tokens. Given SRT's capability to provide finer-grained features (directly applying TokenCut at this level is computationally impractical due to its $O(n^2)$ complexity on constructing a fully-connected graph for graphcut, where $n$ is the number of tokens), we anticipate that future research will develop methods to leverage SRT's high-resolution embeddings effectively.

### 3.5 Sanity Check: Image Retrieval and Unsupervised Object Detection

Incorporating SRT into vision tasks involves updating ViT features based on fine-tuned super-resolved features. However, questions remain regarding whether the observed enhancements in dense prediction tasks are solely due to increased awareness of semantic boundaries in images and whether this method extends to non-dense prediction tasks. To address these concerns, we conducted a sanity check using image retrieval and unsupervised object detection tasks.

For image retrieval, we applied a nearest-neighbor protocol following DINO, using the Oxford image retrieval datasets (Radenović et al., 2018) and ViT-S/16 trained on ImageNet. Notably, our base

| Datasets | ECSSD | | | DUTS | | | DUTS-OMRON | | |
|---|---|---|---|---|---|---|---|---|---|
| Feature Extractor | maxF | IoU | Acc. | maxF | IoU | Acc. | maxF | IoU | Acc. |
| DINO ViT-S/16 | 80.3 | 71.2 | 91.8 | 67.2 | 57.6 | 90.3 | 60.0 | 53.3 | 88.0 |
| +SRT | **82.4** | **71.7** | **92.1** | **68.8** | **58.5** | **90.7** | **61.0** | **54.0** | **88.2** |
| DINO ViT-S/16 w/ bilateral solver | 87.4 | **77.2** | 93.4 | 75.5 | **62.4** | 91.4 | 69.7 | 61.8 | 89.7 |
| +SRT | **88.4** | 77.0 | **93.6** | **76.5** | **62.4** | **91.7** | **70.6** | **62.4** | **89.9** |
| DINO ViT-B/16 | 80.3 | 71.0 | 91.5 | 66.4 | 56.7 | 89.5 | 56.7 | 50.5 | 85.4 |
| + Stochastic Resonance | **81.8** | **72.6** | **92.2** | **68.8** | **58.3** | **90.6** | **58.0** | **51.6** | **86.1** |
| DINO ViT-B/16 w/ bilateral solver | 86.8 | 76.6 | 93.0 | 74.1 | 60.9 | 90.6 | 65.6 | 58.4 | 87.1 |
| + Stochastic Resonance | **88.2** | **78.0** | **93.7** | **68.8** | **58.3** | **90.6** | **67.2** | **59.7** | **87.8** |

Table 3: **Results on unsupervised salient region segmentation.** *Despite architectural constraints, our method yields consistent improvement on all three datasets, with an average increase of 1.8% in the maxF metric.*

| Task | Metric | Baseline | d=1 | d=2 | d=3 | d=4 | d=5 | d=6 |
|---|---|---|---|---|---|---|---|---|
| Image Retrieval | mAP (Medium) | 34.6 | 34.8 | 35.1 | 35.2 | 35.3 | 35.3 | **35.5** |
| | mAP (Hard) | 13.0 | 13.1 | 13.2 | 13.1 | 13.2 | **13.2** | 13.1 |
| Object Discovery | Detection Rate | 68.7 | 68.9 | 68.9 | 69.2 | **69.4** | 69.3 | 69.2 |

Table 4: **Results on Image Retrieval and Object Discovery.** *SRT generalizes to non-dense prediction tasks operating on higher-level region/image features to yield equal or better performance compared to the standard inference baseline. On the Oxford image retrieval task, SRT on the DINO-ViT-S/16 model yields up to 2.6% relative improvement from the baseline model. On the unsupervised object detection task, SRT improves the detection rate by up to 1.0%. d: translation in pixels when ensembling with SRT.*

model's pre-training poses a substantial domain gap to the target datasets. Note that, we do not naively average the class tokens from augmented images, but ensemble the features by SRT prior to the attention mechanism in the last layer. In this way, the final class token is computed from the ensemble SRT feature. Although image retrieval primarily requires distinctive image-level features (rather than pixel-level), aiming to match images to queries at a higher level, SRT exhibited effective adaptation, resulting in a notable 2.6% relative improvement in accuracy.

Regarding unsupervised object detection, we utilized TokenCut and the VOC07 dataset (Everingham et al., 2010). Unsupervised object detection focuses on region-level discriminative features, utilizing bounding boxes instead of segmentation masks for object delineation. Despite this, we observed a 1.0% relative improvement in the detection rate, reaffirming that SRT does not compromise the information within the original ViT embeddings. These results serve as a critical validation of SRT's capacity to enhance ViT features without distorting their original information.

## 4 DISCUSSION

### 4.1 RELATED WORK

**Stochastic Resonance** was proposed by Benzi et al. (1981) and first applied in climate dynamics (Benzi et al., 1982) and later in signal processing (Wellens et al., 2003; Kosko & Mitaim, 2001; Chen et al., 2007) and acoustics (Shu-Yao et al., 2016; Wang et al., 2014). It is used to super-resolve a signal beyond the native resolution of the sensor by adding white noise. We use the same principle to adapt generic ViT image features for dense prediction downstream tasks. By randomly translating the images, (i.e. introducing noise in the spatial dimension), we are able to super-resolve ViT image features to be smoother and better suited for dense prediction tasks. We leave extensions to other groups or semi-groups of transformations (*e.g.*, scale or domain size) to future work.

**Test-time data augmentation** involves aggregating model predictions from augmented test input to a final prediction. Applying such a technique increases the robustness of predictions (Prakash et al., 2018; Song et al., 2017; Cohen et al., 2019) and prediction accuracy (Krizhevsky et al., 2012; Szegedy et al., 2015; Simonyan & Zisserman, 2014; Jin et al., 2018; Matsunaga et al., 2017) in a variety of tasks. It can also used to estimate the uncertainty of the model (Matsunaga et al., 2017; Smith & Gal, 2018; Ayhan & Berens, 2022; Wang et al., 2019). Different transformations are used to target different potential tasks: Pang et al. (2019) linearly combines the testing input and a

randomly sampled clean image to generate classification prediction. Isensee et al. (2018) performs flipping and rotation to the test input image to generate 64 different inputs and finally aggregates the outputs to perform medical image segmentation. Krizhevsky et al. (2012) crops the images into smaller patches and ensemble the results for classification. Self-ensembling (Bousselham et al., 2021) is also closely related to our work. Bousselham et al. (2021) leverages multi-scale features fed into multiple independent decoders to create an ensemble within a single model. Liu et al. (2018) ensembles outputs from networks augmented with random noise layers to improve model robustness. SRT aggregates information via adding spatial translations as noise and can be considered a general case of test-time augmentation, where ensembling is performed at the feature level at intermediate layers of a ViT, instead of the output level, which is novel.

**Knowledge distillation** aims to transfer the knowledge from stronger teacher models to weaker student models to improve their performance. Hinton et al. (2015) trains a student model to mimic the soft output distribution of the teacher model. Romero et al. (2014) extends this idea to distill the intermediate features learned by the teacher models. We consider a form of self-distillation (Zhang et al., 2019), in which the student itself is used as the teacher to improve learned representations.

**Dense ViT feature extractor.** Our work is closely related to (Amir et al., 2021),which employs ViT for generating dense visual descriptors. To extract these fine-grained features, (Amir et al., 2021) reduce the stride allowing for overlapping tokens and performing a single forward pass with ViT. In SRT, instead of a single pass, we conduct multiple passes using perturbed inputs. This modification reduces the computational complexity from quadratic to linear.

## 4.2 CONCLUSION, LIMITATION, AND FUTURE WORK

Averaging sub-token perturbations functions as a versatile visualization tool for ViT embeddings and offers test-time augmentation and ensemble capabilities usable with any ViT architecture. We found the technique especially useful for dense prediction tasks, where even coarse-scale obejcts have fine-scale occlusion boundaries that must be resolved.

In contrast to most test-time augmentation and ensemble methods that operate at the output level, and require task-specific designs, our method can be applied to any layer within any architecture and any task that utilizes ViT as a feature extractor, eliminating the need for modifications to the forward pass, such as increasing token numbers, which would significantly increase GPU usage. Instead, SRT accomplishes this by conducting multiple inferences, avoiding the quadratic complexity related to the number of ViT tokens, and enabling super-resolution of ViT embeddings to match the input resolution. This approach is amenable to parallelization through batching, ensuring computational efficiency. Furthermore, the method allows ensembling without memory-intensive resizing of all embeddings to full resolution, which can be executed recursively, as described in Sect. 2.3. Practical implementations demonstrate efficient execution on even laptop GPUs.

With all that said, Stochastic Resonant Transformers have several limitations. The basic embodiment increases inference cost and latency, as each perturbed image necessitates a ViT forward pass. To address this, one viable approach is knowledge distillation, which involves fine-tuning the network to mimic the feature-level output of SRT. We illustrate this process using the DAVIS-2017 training dataset with DINO-ViT-S/16, achieving improved results (F&J-score $0.617 \Rightarrow 0.625$) without the use of labels or operations on the validation set. This establishes a label-free, task-free transductive fine-tuning scheme that adapts pre-trained ViT features to new target datasets. Future directions may involve refining the distillation process on different layers and exploring the integration of Stochastic Resonance directly into ViT architectures.

Additionally, our findings underscores the segmentation capabilities of ViT embeddings, aligning with recent claims in the field (Caron et al., 2021; Yu et al., 2023). Super-resolved features exhibit sharp, fine-grained semantically relevant boundaries. Furthermore, our method leverages the convexity properties (Park & Kim, 2022) of ViT embeddings, enabling convex combinations (average pooling as a special case) during inference, resulting in improvements across various tasks. It is worth noting that Stochastic Resonance is not limited to ViT architectures nor to spatial quantization. It can be applied to architectures like CNN as well as to other forms of quantization, such as sale or domain size. However, our emphasis in this paper is on ViTs that mostly use non-overlapping tokens, making them particularly suited to our approach.

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

| Architecture | ResNet20 | ResNet32 | ResNet56 |
|---|---|---|---|
| Accuracy | 91.95 | 92.68 | 93.50 |
| Accuracy w/ SRT | **92.41** | **93.14** | **93.87** |
| Relative error reduced | 5.6% | 6.3% | 5.7% |

Table 5: **Results on Cifar-10 classification with ResNet.** *Stochastic resonance consistently improves classification accuracy by an average of 5.87% and as much as 6.3% on ResNet32, without additional training.*

| Method | head | baseline | d=1 | d=2 | d=3 |
|---|---|---|---|---|---|
| DINOV2 ViT-S/14 | linear | 44.24 | 44.44 | 44.57 | **44.64** |
| DINOV2 ViT-B/14 | linear | 47.28 | 47.63 | 47.85 | **47.98** |
| DINOV2 ViT-L/14 | linear | 47.79 | 48.18 | 48.44 | **48.62** |

Table 6: **Results on Semantic Segmentation on ADE20K in mIOU** *Experiments run with evaluation pipeline from InternImage (Wang et al., 2023) and DINOV2 Oquab et al. (2023). d denotes the translation in pixels, ranging from -d to d with respect to a coordinate location across horizontal and vertical directions, when ensembling with SRT. As the size of the ensemble grows, the segmentation mIOU increases.*

## A    RESULTS ON CNNS AND CLASSIFICATION

As mentioned in the paper, Stochastic Resonance is not confined to Vision Transformer architectures. We specifically opted for ViT and zero-shot methods to effectively showcase its benefits. In this context, we present additional results involving Convolutional Neural Networks (CNNs) and supervised image classification. We test ResNet He et al. (2016) on the CIFAR dataset, and report the results in Tab. 5. Through the application of stochastic resonance, we employ ensembling at the final layer before the prediction head. Across ResNet20, ResNet32, and ResNet56, we consistently observe improvements, with the prediction error reduced at inference time by an average of average of 5.87% and as much as 6.3%, all without the need for additional training.

## B    RESULTS ON SEMANTIC SEGMENTATION

We show results on combining semantic segmentation with SRT. We employ the protocol from DINOV2 Oquab et al. (2023) and ADE20K dataset Zhou et al. (2017). The results are presented in Tab. 6. SRT consistently improves mIoU on all three pre-trained ViTs, by as much as 1.7% in relative improvement. In comparison, results on depth prediction show a more significant improvement (14.9%). We conjecture that depth prediction benefits more from SRT as it is a geometry task and SRT leverages a geometric augmentation. Nevertheless, the gain in semantic segmentation is obtained without any further fine-tuning.

## C    DISCUSSION: COMPARISON WITH FEATURE INTERPOLATION

One might speculate that the performance improvement of SRT arises from a fine-grained feature map, which may be advantageous for dense prediction tasks. We conduct a sanity check in Sect. 3.5 in the main paper, and here we further compare SRT with resizing the feature field through interpolation. We adopt the depth prediction task using NYU-V2, consistent with the main paper, and the results are presented in Tab. 7. Somewhat surprisingly, the simple approach of interpolating the feature map leads to a performance decrease. Two possible explanations are considered. First, the architecture and training loss lack an explicit constraint on the smoothness of features, making spatial interpolation problematic. Second, while interpolation increases feature resolution, it does not introduce new information, whereas ensembling effectively samples the input signal (augmented image) multiple times, which contains more information than any single sample.

Considering the ensembled features represent a "denoised" signal, we can measure the noise distribution in the features by the L2 difference between the ensembled features and the feature from a single forward pass, aggregated as a histogram. Fig. 4 visualizes this noise distribution. It is easy to notice that, the noise is mostly aligned to the semantic boundaries where image patches do not align with object shape due to quantization. In comparison, we also visualize the difference between the

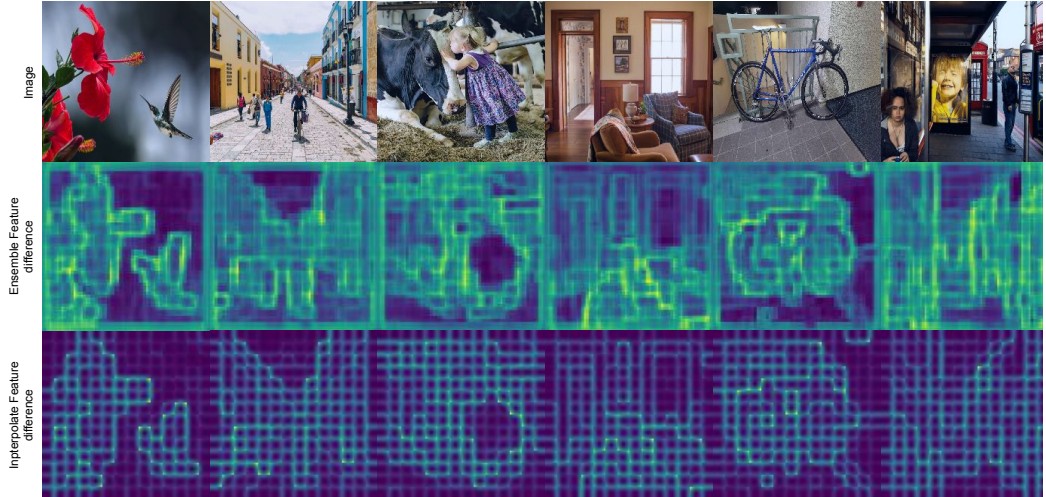

Figure 4: **Noise distribution in the features by SRT.** *Considering the ensembled features represent a "denoised" signal, we visualize noise distribution, which aligns to semantic boundaries (2nd row) where image patches do not align with object shape due to quantization. For reference, we also show the difference between resized features (by interpolation) to the original feature, which shows a less meaningful grid pattern.*

| Backbone | Head | Method | RMSE | RMSE_log | AbsRel | SqRel | a1 | a2 | a3 |
|---|---|---|---|---|---|---|---|---|---|
| DINOV2-ViT-S/14 | DPT | Baseline | 0.336 | 0.114 | 0.080 | 0.048 | 0.933 | 0.986 | 0.996 |
| | | Bilinear | 0.573 | 0.178 | 0.146 | 0.125 | 0.8 | 0.964 | 0.995 |
| | | Bicubic | 0.146 | 0.124 | 0.572 | 0.178 | 0.801 | 0.964 | 0.995 |
| DINOV2-ViT-B/14 | DPT | Baseline | 0.323 | 0.109 | 0.074 | 0.044 | 0.941 | 0.987 | 0.996 |
| | | Bilinear | 0.568 | 0.185 | 0.146 | 0.12 | 0.792 | 0.96 | 0.992 |
| | | Bicubic | 0.579 | 0.188 | 0.149 | 0.124 | 0.787 | 0.959 | 0.991 |
| DINOV2-ViT-L/14 | DPT | Baseline | 0.311 | 0.105 | 0.070 | 0.042 | 0.946 | 0.988 | 0.997 |
| | | Bilinear | 0.732 | 0.246 | 0.183 | 0.195 | 0.695 | 0.91 | 0.973 |
| | | Bicubic | 0.720 | 0.241 | 0.181 | 0.188 | 0.701 | 0.916 | 0.975 |

Table 7: **Results on NYU-V2 depth prediction using interpolated features** *For the interpolation method, we bilinearly or bicubically interpolate the DINOV2 feature up to the image dimension and perform an average pooling to return the feature to the original dimension. The results are much worse than the baseline method, which uses the original DINOV2 features without ensembling.*

resized features (by interpolation) and the single-forward-pass feature. The resulting noise map is a less meaningful grid-like structure due to the image quantization and interpolation artifacts. Note that the center of each patch of the difference maps of the interpolated features is "dark" meaning there is no information introduced. On the contrary, the difference maps of SRT is "bright", which comes from the noise introduced by Stochastic Resonance to enhance the signal.

## D    PSEUDO-CODE FOR SRT

The pseudo-code for SRT is provided in Alg. 1.

## E    FORMAL RELATION BETWEEN SRT-INDUCED REPRESENTATIONS AND EXPLICIT MODELS OF IMAGE FORMATION

Stochastic Resonance is a phenomenon whereby *"increases in levels of unpredictable fluctuations – e.g., random noise – cause an increase in a metric of the quality of signal transmission or detection performance, rather than a decrease "* (McDonnell & Abbott, 2009) The phenomenon was first ana-

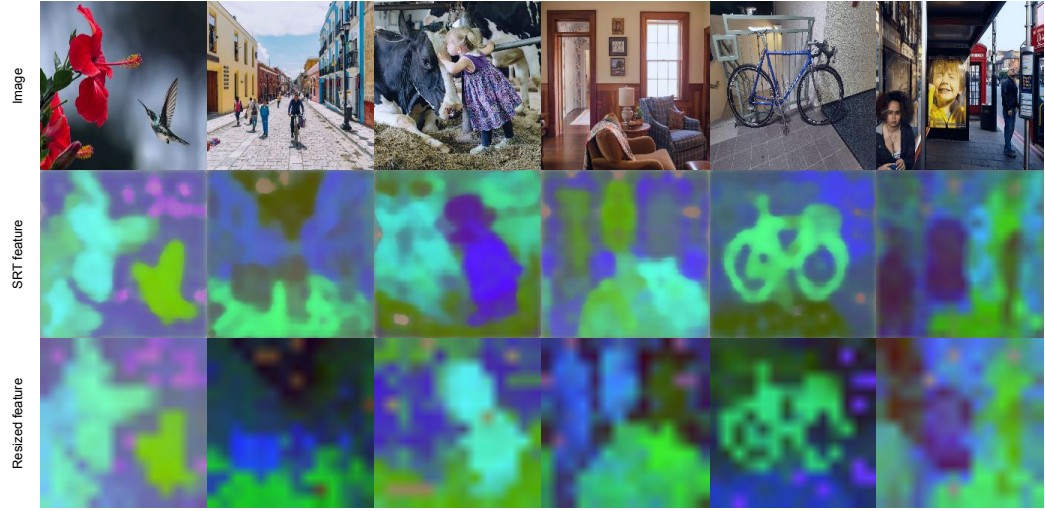

Figure 5: **Comparing SRT features and resized single-forward-pass features.** *SRT features respect the semantic boundaries better than resized features (The edge of the bird and pedals in column one and wheels of the bicycle in column five). Resized features contain quantization artifacts where edges are vertical and horizontal lines corners are right angle corners. Our feature can represent much more detailed object contours.*

---

**Algorithm 1** ALGORITHM TO OBTAIN SRT FEATURES.

---

**Require:** ViT feature extractor $f_\theta$, Image $I \in \mathbb{R}^{3 \times H \times W}$, Level of pertubation $d \in \mathbb{N}_+$
1: $b \leftarrow \{0\}^{N \times H \times W}$, where $N$ is the dimension of the ViT features.
2: **for** $x \in \{-d, \dots, 0, \dots, d\}$ **do**
3:     **for** $y \in \{-d, \dots, 0, \dots, d\}$ **do**
4:         Translate $I$ by $x$ and $y$ pixels in the horizontal and vertical direction respectively to get $I'$
5:         Obtained the ViT feature for $I'$: $f_\theta(I')$
6:         Upsample $f_\theta(I')$ to image dimension $H \times W$ to obtain $f'_\theta(I') \in \mathbb{R}^{N \times H \times W}$
7:         Translate $f'_\theta(I')$ by $-x$ and $-y$ in the horizontal and vertical direction respectively to obtain $f''_\theta(I') \in \mathbb{R}^{N \times H \times W}$
8:         Aggregate $f''_\theta(I')$ into $b$
9:     **end for**
10: **end for**
11: **return** $\frac{b}{n_b}$, where $n_b$ is the number of features aggregated into $b$

---

lyzed in reference to bi-stable systems of stochastic differential equations, where the variance of the observations decreased after injecting noise into the system (Benzi et al., 1981). Subsequently, the idea was applied to quantized systems, where each level $x^i \in X = \{x^1, \dots x^N\}$ of a quantization operator $f : \mathbb{R} \to X \subset \mathbb{R}$ replaced the role of each stable attractor in the original formulation, and stochastic resonance resorted to averaging noisy qantized versions of the original signal $x$,

$$\hat{x} = \int f(x + n) dP(n) \tag{5}$$

resulting in smaller noise variance $\mathbb{E}(|\hat{x} - x|^2)$ (or higher signal-to-noise ratio) than the original measured signal without added noise, $\mathbb{E}(|x - f(x)|^2)$ where $\tilde{x} = f(x)$, all assuming sufficiently small perturbations $n \sim dP$ from a chosen distribution $P$ (a design choice). This method was successfully used in cochlear signal processing where $f$ is a coarse quantizer implemented using low-power electronics.

Now consider the more general case where $x$ is not just a real number but an image (irradiance function) defined on a compact planar domain, $x : D \subset \mathbb{R}^2 \to \mathbb{R}; (u, v) \mapsto x(u, v); f$ is a complex operator that, in addition to quantizing the planar domain $D$ with a function $\pi : D \to \Lambda$, where $\Lambda$ is

a discrete lattice, also maps each cell in the lattice in a $K$-dimensional vector $\phi(\{x(u,v),(u,v) \in D_i\}) \in \mathbb{R}^K$, where $D_i$ is a cell in the lattice, so that $f : D \to \mathbb{R}^K; x \mapsto f(x)$ with

$$f(x) = \phi(\pi(x)) \tag{6}$$

where $\pi$ denotes the restriction to the lattice. Now, instead of considering an additive perturbation that acts on the range space of $x$, $x \mapsto x + n$, we consider a more general perturbation operator $T$ that can act on either the domain $D$ or the range $x$, which we indicate with $Tx$. This can be a trivial additive perturbation, $T(n)x = x + n$ for some scalar $n$, or it can be a planar translation $T(n)x(u,v) = x(n + n_u, v + n_v)$, where $n = (n_u, n_v)$ is a translational offset, or it can be an affine, projective, diffeomorphic or homeomorphic deformation of both the domain and the range of $x$. In this paper, we restrict ourselves to planar translation but the concepts extend to any invertible operator $T$. We write this formally as

$$\hat{x} = \int f(Tx(u,v))dP(T(u,v))$$

which boils down to spatial averaging if we choose $dP$ to be constant ($P$ uniform). One can also consider scale averaging, which gives rise to so-called domain-size pooling (Dong & Soatto, 2015). In a discrete setting, the perturbation can be quantized and averaged

$$\hat{x} = \sum_i \phi(\pi(x_i))$$

which can done sequentially as a moving average. Our model includes a slightly more sophisticated (forward-backward) projection operator $\pi$, that allows us to express the averaging in terms of the measured signal rather than the (unknown) original analog signal, since the latter is unknown.

Now, in addition to reducing the variance of the reconstruction error, what are the specific artifacts that arise in the presence of spatial quantization that we wish to recuperate by the use of perturbation-averaging?

Consider an elementary image-formation model where a physical scene is composed of piecewise smooth multiply-connected surfaces, $S \subset \mathbb{R}^3$, each supporting a piecewise smooth radiance function ("texture" or albedo) $\rho : S \to \mathbb{R}$, imaged through a pinhole (central) projection $\pi : \mathbb{R}^3 \to \mathbb{R}^2$, where the projection operator also includes spatial quantization into the lattice. Now, we have:

$$\tilde{x}(\tilde{u}, \tilde{v}) = \int_{\pi^{-1}(\tilde{u},\tilde{v})} \int_L \beta_{p(\tilde{u},\tilde{v})}(\tilde{u} - u, \tilde{v} - v)dE(u,v); \quad [\tilde{u}, \tilde{v}] = \pi(p),\ p \in S$$

where $\beta$ is a bi-directional reflectance distribution function (whose integral over a unit solid angle around $(u,v)$ at each point $p$ yields the diffuse albedo $\rho$) and the integral over the light source $L$ extends to the pre-image under $\pi$ of the quantization cell around $(\tilde{u}, \tilde{v})$ (this is the intersection of the cone subtending a patch centered at $(\tilde{u}, \tilde{v})$ with reflective surfaces in the scene). Notice the inverse projection in the domain of integration and the forward projection in the selection of the projection point, corresponding to a cell in the lattice.

One should also notice that the representation $\phi(x)$ computed at location $(\tilde{u}, \tilde{v})$ does not just use the pixel at that location, nor does it simply average pixels in its neighborhood, but rather aggregates information from all pixels in the patch. Nonetheless, the information about the *scene* that is being aggregated changes with the distance of the scene, and translating the patch (*e.g.,* computing the representation at an adjacent patch, even if overlapping) mixes the contribution of different connected components of the underlying surface $S$, and corresponding segments of the albedo $\rho$. Therefore, even if vectorial, the representation $\phi$ is subject to quantization artifacts *when viewed as a representation of the scene, rather than of the given patch*, which causes artifacts such as the loss of details at occluding boundaries and albedo boundaries. By aggregating multiple samples at different values of the transformation $T$, we can recuperate some of those details, up to the quantization limits of the sampling of the transformation (as opposed to the quantization limits of the patch-based tokenization).

Generally, the quantized signal $\tilde{x}$ is piecewise constant, but the discontinuities correspond to the lattice cell boundaries, and have nothing to do with either the geometric discontinuities due to the piecewise smooth nature of $S$, or the photometric discontinuities due to albedo boundaries in $\beta$ or

the corresponding $\rho$. As a result, object boundaries (which correspond either to occlusion/geometric boundaries, or material/albedo boundaries) are not visible in the quantized signal and generally can appear and disappear even at constant quantization levels simply by moving farther and closer to the camera due to the varying size of the intersection of the cone $\pi^{-1}(D_i) \cap S$. This gives rise to genetic effects of the kind familiar in scale space theory (Lindeberg, 2013).

While it would be ideal to be able to prove analytically that discontinuities due to material and illumination boundaries that are washed out by spatial quantization are recovered by stochastic resonance, even the simplistic image formation model above is way beyond the complexity that is amenable for direct analysis. For this reason, in the paper we resort to empirical tests, either qualitative by direct visualization, or qualitative by using the averaged feature $\hat{x}$ instead of the original feature $\tilde{x}$, in downstream inference tasks.

## F  ADDITIONAL VISUALIZATION

We offer additional visualization of ensembled SRT features across various network layers, using CLIP and DINO for illustration. Our visualization indicates a noticeable trend: deeper layers reveal clearer high-level semantic boundaries, while shallower layers highlight more local features compared to high-level ones.

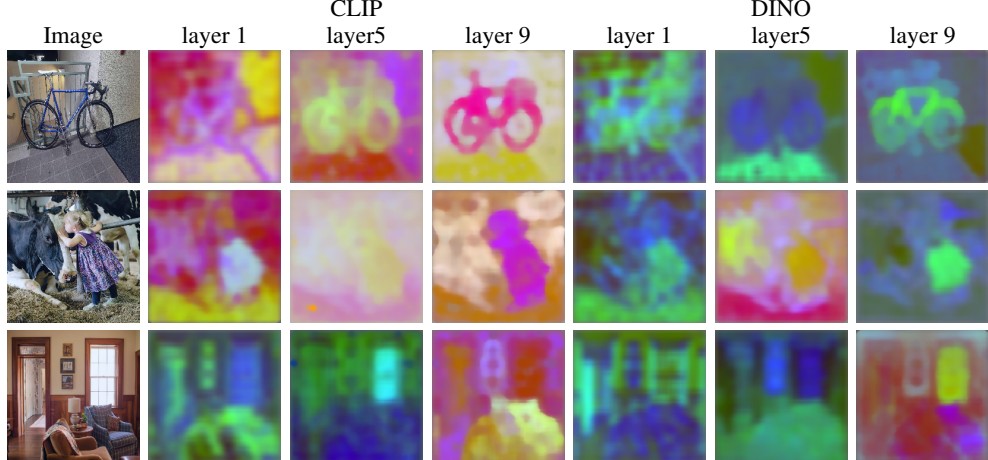

Figure 6: **Visualization of ensembled SRT features in different ViT layers.** *Architecture: ViT-S/16.*

