# OpenReview forum: "Sub-token ViT Embedding via Stochastic Resonance Transformers"
_ICLR.cc/2024/Conference — Submitted to ICLR 2024_

### Official Review · Reviewer_3VWw · 2023-10-18

**Soundness:** 4 excellent
**Presentation:** 3 good
**Contribution:** 3 good
**Rating:** 8
**Confidence:** 4

**Summary:**

This paper proposes a "Stochastic Resonance Transformer" (SRT) method that improves the performance of pre-trained Vision Transformers (ViTs) in downstream tasks. SRT achieves this by applying controlled perturbations to input images (i.e., sub-token spatial translations) and super-resolve features of pre-trained ViTs, capturing more of the local fine-grained structures that might be neglected by tokenization.

**Strengths:**

Clarity of Presentation: The paper is well-written and easily understandable. It effectively conveys the proposed approach and its rationale.
Simplicity and Effectiveness: The simplicity of SRT is a notable strength. Despite its simplicity, it has demonstrated high effectiveness in five vision tasks, which is a valuable contribution to the field.
Generalization Ability: SRT can be applied at any layer and on any task without fine-tuning.

**Weaknesses:**

The theoretical guarantee is missing

**Questions:**

1. Can SRT be applied to models beyond Transformers? It would be of interest to see empirical results exploring its applicability to other models such as ResNets or MLPs.
2. It is important to understand the hyper-parameters involved and perform in-depth analysis when noise level can be effective or harmful. Since stochastic resonance may not be suitable for all cases, how to determine the optimal noise levels in practical applications?
3. It would be helpful to analyze the noise distribution and perform experiments using other augmentation strategies.
4. Why not conduct experiments on the main task in computer vision such as image classification, semantic segmentation, and object detection?

---

> ### Author Response · Authors · 2023-11-16
> **Author response to the initial review**
>
> **Q1:** *Can SRT be applied to models beyond Transformers? It would be of interest to see empirical results exploring its applicability to other models such as ResNets or MLPs.*
>
> **R1:** As stated in the paper, 'It is worth noting that Stochastic Resonance is not limited to ViT architectures nor to spatial quantization. It can be applied to architectures like CNN as well as to other forms of quantization, such as sale or domain size. However, our emphasis in this paper is on ViTs that mostly use non-overlapping tokens, making them particularly suited to our approach.' (Section 4.2)
>
> As suggested by the reviewer, we applied SRT to ResNet on classification problem on Cifar, the results are in the Appendix A, where we improve classification accuracy by an average of 5.87%.
>
> **Supervised classification on Cifar-10**
> Architecture | ResNet20 | ResNet32 | ResNet56
> | ----------- | ----------- | ----------- | ----------- |
> | Accuracy | 91.95 | 92.68 | 93.50 |
> | Accuracy w/ SRT | **92.41** | **93.14** | **93.87** |
> | Relative Error Reduced | 5.6% | 6.3% | 5.7% |
>
> **Q2:** *It is important to understand the hyper-parameters involved and perform in-depth analysis when noise level can be effective or harmful. Since stochastic resonance may not be suitable for all cases, how to determine the optimal noise levels in practical applications?*
>
> **R2:** That's a great question, and admittedly an ongoing effort to study how to best determine them outside of empirical trials. In our experiments, we noticed that using an augmentation level close to one-fourth of the patch size works effectively for various tasks. We look forward to future research shedding light on this question.
>
> **Q3:** *It would be helpful to analyze the noise distribution and perform experiments using other augmentation strategies.*
>
> **R3:** We appreciate the reviewer's input but may need further clarification on the term 'noise distribution' that the reviewer is referring to. If the reference is to artificial noise added as augmentation for stochastic resonance, then indeed, various options are available, as long as they are group transformations which guarantees an inverse transformation. However, we specifically chose translation, as it avoids introducing interpolation artifacts, unlike rotation, and provides finer control over the ensemble (pixel displacement in rotation depends on its distance from the center). Importantly, for transformers with positional encodings, flipping is not a viable option since the embeddings are not designed to be invariant.
>
> If the term 'noise' pertains to stochastic noise in the features, assuming the ensembled features represent a "denoised" signal, we measured it by the Euclidean distance between the ensembled feature and the feature from a single forward pass, aggregated as a histogram. We show this noise distribution in Fig. 4 in the revised manuscript's Appendix.
>
> **Q4:** *Why not conduct experiments on the main task in computer vision such as image classification, semantic segmentation, and object detection?*
>
> **R4:** We focus on zero-shot tasks to highlight the capabilities of SRT without any fine-tuning. However, as explained in the paper, SRT can be used for any layer in any Vision Transformer (VIT). In response to the suggestion, in addition to the classification results presented in Q1, we present results for semantic segmentation in the Appendix B. SRT consistently improves mIoU on all three pre-trained ViTs, by as much as 1.7% in relative improvement. A discussion is provided in the Appendix:
>
> **Semantic segmentation by DinoV2 with linear head on ADE20K**
> Architecture | baseline | d=1 | d=2 | d=3
> | ----------- | ----------- | ----------- | ----------- | ----------- |
> | ViT-S/14 | 44.24 | 44.44 | 44.57 | **44.64** |
> | ViT-B/14 | 47.28 | 47.63 | 47.85 | **47.98** |
> | ViT-L/14 | 47.49 | 48/18 | 48.44 | **48.62** |

---

> > ### Comment · Reviewer_3VWw · 2023-11-23
> > **Reply to the authors**
> >
> > By summaring the comments from other reveiwers and the reply from the authors, we have the following additional question.
> > 1. The problem statement regarding the exact nature of these artifacts lacks clarity in its definition. Please provide an in-depth characterization and analysis of the artifacts present in feature maps. For instance, it is crucial to determine which specific layers of the transformer these artifacts typically appear in, at which training stage they tend to manifest, and whether there is any correlation between the occurrence of artifacts and the size of the model.
> > 2. There have been other works [R1] developed for improving artifacts in the feature extraction process with pretrained models. Please analyze the relevance and differences between these works and the present study.
> > [R1] Darcet T, Oquab M, Mairal J, et al. Vision Transformers Need Registers[J]. arXiv preprint arXiv:2309.16588, 2023.

---

> ### Author Response · Authors · 2023-11-23
> **Nature of the artifacts, additional related work**
>
> Thank you for bringing up the questions.
>
> **1.** To delve further into the 'exact nature,' we have included an additional theoretical analysis in Appendix E, please also see our response to Reviewer wDbQ. The artifact we're tackling is related to tokenization, so it should be present in every layer of ViT. However, as the features go deeper, they contain more high-level information and the artifact potentially affects the downstream task more significantly. We are delighted to add extra visualizations for the feature map at different layers in Figure 7 to bring more insights to the question.
>
> **2.** Reference [R1] is a concurrent ICLR submission (can be found by paper title search), and its Arxiv version was released after the paper submission deadline, so we could not discuss it in our initial manuscript. However, we're happy to offer additional insights:
>
> [R1] describes a phenomenon that when trained on specific tasks (such as contrastive, visual-language, etc.), some tokens in the "image background", considered "uninformative" for the pre-training task, get repurposed, possibly incurring decreased performance on dense prediction tasks. The proposed solution involves adding extra tokens to compensate for the repurposed tokens, ensuring that local ViT tokens represent local features of the image.
>
> Although what our work proposes is an ensembling method, which differs from [R1], two papers are indeed relevant. Please see the upper left side of Figure 1 in our paper. When CLIP is applied to the bicycle image, there are purple 'blobs' in the background, possibly corresponding to the 'repurposed tokens' described by [R1]. Instead of adding tokens, SRT reduces this artifact through ensembling. In the ensembled feature map (labeled as 'ours'), these purple blobs are smoothed out and eliminated. [R1] therefore provides a potential explanation for why SRT shows a significant improvement in performance for dense prediction tasks like depth prediction. Thanks to the reviewer for bringing this paper up and we're open to incorporating a detailed discussion in our manuscript.
>
> We hope this response answers the reviewer's questions. If there are further concerns, we are more than willing to provide additional explanations.

---

### Official Review · Reviewer_j1cP · 2023-10-31

**Soundness:** 2 fair
**Presentation:** 2 fair
**Contribution:** 2 fair
**Rating:** 5
**Confidence:** 3

**Summary:**

In this work, a sub-token VIT embedding based method is proposed to increase the resolution of the intermediate feature maps from VIT. The method is called Stochastic Resonance Transformer (SRT). A set of perturbations(only translations for now) are applied to the input image. These set of perturbed images are then fed into VIT to get a series of token embeddings. These features are then upsampled to higher resolution and aligned using the inverse of the applied perturbations. Statistical aggregation, including mean and median, along the perturbation dimension, produces fine-grained feature representations. This method is evaluated on multiple CV tasks like object segmentation, depth estimation and unsupervised saliency segmentation.

**Strengths:**

- This work introduce an interesting finding that super-resolution on the token embedding can help improve the VIT's capability
- The effectiveness of this method has been verified in multiple CV tasks.

**Weaknesses:**

- Perturbations are translation only. Translations are one of the perturbations, and can be replaced by a simple convolution kernel. If translation based perturbation works, then it's possible that other complex perturbations like rotation should also work. Also, since the translation works, it seems it can be replaced by an convolution network, followed by some deconvolution kernels in the upsampling stage, with some conv layers to do the aggregation. Then it will become a learnable model instead of translations only. In this way the model can not only simulate the translation, but also other non-linear perturbations.
- Did not compare with image super-resolution method. Image super resolution is an well-studied area, and it's an natural thought to try some of them and see if the generated super-resolution features can help to improve the down-stream tasks.

**Questions:**

- Is it possible to try other image super resolution models and see how the performance looks like? Theoretically, if the simple resizing works, then other super resolution should work better.
- Is it possible to use some convolution kernels to replace the predefined perturbation type?

---

> ### Author Response · Authors · 2023-11-16
> **Author response to the initial review**
>
> **W1:** *Perturbations are translation only. Translations are one of the perturbations, and can be replaced by a simple convolution kernel. If translation based perturbation works, then it's possible that other complex perturbations like rotation should also work. Also, since the translation works, it seems it can be replaced by an convolution network, followed by some deconvolution kernels in the upsampling stage, with some conv layers to do the aggregation. Then it will become a learnable model instead of translations only. In this way the model can not only simulate the translation, but also other non-linear perturbations.*
>
> **R1:** The reviewer is correct that we could replace the translated pillbox kernel we use (although we do not explicitly mention) with any other kernel undergoing any other group (invertible) transformation. Those would give rise to generalized forms of convolution such as Fourier-Mellin etc., which indeed could further improve our method if the sampling of the group is adapted to the signal. We simply use the most obvious choice (pillbox = constant) of kernel and of group (translation) for simplicity.
>
> Moreover, from an engineering perspective, employing translation as a group transformation offers several advantages:
> Assuming pixel displacements in the translation matrix, one can map every feature to discrete pixel locations, which avoids interpolation artifacts commonly associated with other transformations, i.e., rotation, scaling.
> Translation does not change the size for objects in the image, resulting in more stable features than scaling.
> Unlike rotation, where pixel displacement varies based on distance from the center, pixel displacement of each patch can be directly controlled by specifying translation components in the matrix.
> We appreciate the reviewer's valuable suggestions for future work. However, it's essential to note that in this study, translation performs well and is validated by improvement in five different tasks.
>
>
> **W2:** *Did not compare with image super-resolution method. Image super resolution is an well-studied area, and it's an natural thought to try some of them and see if the generated super-resolution features can help to improve the down-stream tasks.*
>
>
> **R2:** (Please also see the "general message".) We wish to emphasize that ours is an ensembling method, not a super-resolution method in the traditional sense: We capture multiple samples (feature maps) from the original signal (image), which span a sigma-algebra strictly containing that of a single sample, from which one cannot retrieve the information lost simply by post-processing. We could use an analogy with acoustic processing where super-resolution would be akin to noise reduction, whereas stochastic resonance is threshold reduction. There may be a misunderstanding in the term “super-resolution”, in the sense that we do not increase the size, but enhance the detail. We will revise the text to make this clear.
>
>
> **Q1:** *Is it possible to try other image super resolution models and see how the performance looks like? Theoretically, if the simple resizing works, then other super resolution should work better.*
>
> **R3:** Regarding super-resolution, please see the [general message].
>
> In practice, the use of image super-resolution on the features is not feasible because, unlike the standard 3 channels (RGB), low-resolution features typically have many more channels (e.g., 256 for ViT/S or 768 for ViT/B). Existing super-resolution methods are not designed to handle inputs with such a high number of channels. We acknowledge the reviewer's concern and conducted experiments by resizing, some results are highlighted below (for a full table, see Table 7 in the revised paper). Overall simple resizing is detrimental to the task. This may be because there is no spatial smoothness constraint that is imposed on ViT architecture or training process, so a simple resizing operation introduces out-of-distribution features to the network.
>
> **Feature interpolation on monocular depth estimation with NYU-V2 and DinoV2**
>
> Architecture | Method | RMSE | RMSE_log | AbsRel | SqRel
>  ----------- | ----------- | ----------- | ----------- | ----------- | -----------
> | ViT-S/14 | baseline | **0.336** | **0.114** | **0.080** | **0.048** |
> | | interpolation | 0.573 | 0.178 | 0.146 | 0.125 |
>
>
> **Q4:** *Is it possible to use some convolution kernels to replace the predefined perturbation type?*
>
> **R3:** In theory, perturbations are not restricted, as long as they adhere to group transformations, and translation can indeed be defined using convolution. However, in practice, finding more effective kernels remains an open problem, requiring further research. This paper concentrates on zero-shot tasks, where no additional training is performed. Nevertheless, we acknowledge that learned augmentations (so long as they are invertible) could be an interesting idea for certain specific tasks.

---

> > ### Comment · Reviewer_j1cP · 2023-11-22
> > **response to rebuttal**
> >
> > Thanks for the rebuttal. It's surprising to see the simple resizing is not working here. Questions
> > - What's the interpolation method used here? The most common one is the bicubic one.
> > - For model based super-resolution solution, it's still possible to apply to the feature maps here, we can either train a gray-scale image super-resolution model (what's used in medical image super-resolution), or do super-resolution with each feature map (copy to RGB channels)
> > At this time I tend to keep my original rating

---

> ### Author Response · Authors · 2023-11-22
> **Interpolation, super-resolution**
>
> **Q:** *What's the interpolation method used here? The most common one is the bicubic one.*
>
> **R:** We are reporting results using bilinear interpolation. As suggested, bicubic interpolation also yields following similar inferior results as bilinear interpolation, as shown by the following table:
>
> Architecture | Method | RMSE | RMSE_log | AbsRel | SqRel
>  ----------- | ----------- | ----------- | ----------- | ----------- | -----------
> | ViT-S/14 | baseline | **0.336** | **0.114** | **0.080** | **0.048** |
> | | Bilinear interpolation | 0.573 | 0.178 | 0.146 | 0.125 |
> | | Bicubic interpolation | 0.572 | 0.178 | 0.146 | 0.124 |
>
> Please also see Fig. 4 in the updated Appendix for a detailed discussion.
>
> **Q:** *For model based super-resolution solution, it's still possible to apply to the feature maps here, we can either train a gray-scale image super-resolution model (what's used in medical image super-resolution), or do super-resolution with each feature map (copy to RGB channels) At this time I tend to keep my original rating*
>
> **R:** If the reviewer is suggesting, for a $[h,w,768]$ feature map, map each $h\times w\times 1$ channel to a gray-scale image, run through a super-resolution network, and output a fine-grained feature, e.g. $[h\times 16, w\times 16, 768]$, this is infeasible since:
> Super-resolution (SR) networks are domain-specific, hence applying SR networks pre-trained on, for instance, RGB/gray-scale natural images, would generalize poorly to high-dimensional feature maps. Indeed, as the reviewer rightfully pointed out, a proper comparison would require training a super-resolution network for each dimension of the feature map. This is well beyond the scope of our paper and comparisons since our method is zero-shot and does not require any training.
>
> Please also refer to *A general message to the reviewers and AC* for a detailed discussion about super-resolution. If there is related work that we are currently unaware of, we would appreciate it if the reviewer could point to the exact literature, so that we can compare/respond accordingly.

---

### Official Review · Reviewer_wDbQ · 2023-11-02

**Soundness:** 3 good
**Presentation:** 3 good
**Contribution:** 2 fair
**Rating:** 5
**Confidence:** 5

**Summary:**

The paper presents the problem of quantisation artefacts in ViT features as a byproduct of the standard tokenisation procedure of partitioning images into non-overlapping tokens. As a remedy, the authors propose resolving patch contributions by perturbing the input by stochastic translation of the input image and subsequent aggregation in the embedding space to extract sub-token resolution features.

The authors directly refer to their method as a form of stochastic resonance. In classical stochastic resonance, thresholded boundaries are softened through the addition of stochastic noise, and the idea the paper seems to propose is that discrete boundaries of the partitioning can be super-resolved via stochastic perturbation with translations to allow neighbouring tokens to share information when aggregated. The goal seems to be to preserve finer grained spatial details and hence reduce the impact of the general partitioning in classical ViT tokenisation. The approach is reminiscent of classical dithering. The paper also mentions that the approach could easily be extended to other perturbations or augmentations, and is not necessarily

Applications of the method is demonstrated, but largely limited to post-hoc or few-shot modelling approaches where features are further processed for dense prediction tasks; including upscaled feature visualisations using PCA, experiments on video object segmentation, and monocular depth prediction. Two non-dense downstream tasks are also demonstrated.

In summary, this reviewer find certain parts of the paper interesting and novel, particularly as a ensembling distillation method. However the methodology is murky at points, and the presented problem statement and applications somewhat undersells what this reviewer consider to be the strong points of the method.

**Strengths:**

- The proposed method is intuitive and based on well-established principles in signal processing, while being remarkably simple to apply, similar in form to an augmentation and ensembling scheme.
- The method is essentially post-hoc and architecture agnostic as it only requires super-resolving features extracted the final layer, which could then be leveraged in potential downstream tasks, notably dense prediction tasks which require higher resolution feature attributions.
- Aside from the outlined method using translation for super-resolving features, the method itself could be formulated as general augmentation-ensembling method, and is touched upon in the paper. This further makes the method potentially interesting for self-supervision and fine tuning.
- The authors outline a distillation scheme that improves performance in certain downstream tasks. This posits the method as similar to augmentation for an ensemble based fine tuning tasks, which seems apt for further investigation. This distillation process could potentially be used in improved fine tuning for general ViT modelling approaches.
- The paper seems to want to discuss what this reviewer considers an important and often overlooked limitation inherent in the canonical ViT architecture, where uniform partitioning might not align well with the spatial semantic content in the image.

**Weaknesses:**

- ~~The problem the paper seeks to tackle, while somewhat intuitively reasoned, seems insufficiently motivated. The quantisation artefacts are presented as a result of discrete partitioning into uniform square patches, but the exact nature of these artefacts as well as their effect on predictions are hardly discussed or touched upon. This makes the problem statement more ambiguous than necessary.~~ **Edit:** *The last revision addresses this with a formalisation of the context of SRT.*
- While the upscaled feature visualisations show improvement on the low resolution visualisations of the base model, ~~simpler interpolation methods exist for this express purpose. It is also not clear if these new feature maps can be said to be faithful as interpretations of the importance of the features with regard to the predictions.~~ **Edit**: *The authors expand their experiments with alternative interpolation methods, however the main concern on how useful the visualisations are as attributions for model predictions is not clear. The authors revise the manuscript and does no longer claim this as one of the main contributions of the work.*
- ~~While the authors mention the conceptual overlap with Amir et al. (2021), this connection is not touched on to any further extent in the paper, except for a discussion on computational complexity. It seems natural to contrast against the results in this work since, by the authors own admission, there is an inherent similarity between approaches.~~ *The authors expands the link in their revised work with additional results.*
- Methodology for non-dense downstream tasks (particularly image retrieval) seems unclear, and hardly reproducible. Several questions remain unanswered for practitioners wanting to reproduce the results in the paper. ~~Additionally, there seems to be little to no mention of input resolutions used for the modelling.~~ **Edit:** *Note on resolution was addressed. While the authors seek to address gaps in methodology by releasing their code, more exposition in the manuscript would be ideal, but is further addressed in the last revision*.

We also detail some minor weaknesses:
- ~~The link to stochastic resonance, while somewhat clear to this reviewer, is a little opaque. While this analogy could provide an intuitive understanding of the method, it is imperative to clarify the link and show the extent to which quantisation artefacts inhibit ViTs. Stochastic resonance typically involves the enhancement of weak signals in the presence of noise. In this context, techniques are applied for extracting super-resolved features by treating the discrete boundaries imposed by the partitioning in the tokeniser as the thresholds. In this reviewers opinion, the link should be emphasised and clarified to improve the overall contribution of the paper.~~ *The authors agree that this is an issue, and have substantially revised their manuscript.*
- While the authors highlight the computational limitations of ensembling, at times the wording in the article seems to point to computational benefits, e.g. "Practical implementations demonstrate efficient execution on even laptop GPUs". ~~At the same time, the study is limited to ViT-S16 capacities due to computational considerations. While we wholeheartedly agree that not all studies need to be extrapolated to models with huge memory footprints, the limitation to small architectures strikes the author as a little too restrictive, and using a base model (ViT-B16) would have been more convincing.~~ *The authors included larger models in subtasks and expand their results for segmentation in the revised paper.*
- While the method is novel, the current applications seem moderately niched.

**Summary from rebuttal:** *Several of the concerns were addressed in the last revision, and while certain doubts still linger on the potential impact of the method given the role of new tokenisation methods to alleviate quantisation artefacts, the method is novel with promising initial results.*

**Questions:**

- ~~What is the precise nature of the artefacts the authors wish to remedy? Are the authors arguing that the low resolution PCA / attention maps exhibits these artefacts by virtue of having low resolution?~~ *This was addressed in the last revision.*
- Have the authors considered adding metrics for evaluating the faithfulness of the attributions in the visualisations of the features? **Edit:** *The authors downplay the contribution of the visualisations in their contributions, addressing this concern.*
- ~~Why were contrastive comparisons to Amir et al. (2021) (in terms of results) omitted?~~ *The authors diligently expand the discussion and experiments in the revision. The exposition from the rebuttal could be meaningfully be appended to the manuscript for a more nuanced read.*
- ~~Exactly which embeddings were used for the KNN in the image retrieval task? The ensemble of class tokens? Pooled features? This is very unclear.~~ *The authors address this in their revision, however the methodology for non-dense prediction tasks is still unclear.*
- ~~The metrics for results in table 4, are they mAP? What does the columns 1-6 refer to?~~ *The authors address this in the revision.*

An additional question was added in discussion:
- ~~The scores with salient segmentation using TokenCut omit comparisons with post processing using the bilateral solver proposed in the original paper. The reasoning behind this choice seems unclear [...]~~ *The authors addressed this by expanding their results.*

---

> ### Comment · Reviewer_wDbQ · 2023-11-15
> **On salient region segmentation**
>
> Some additional concerns; in reviewing the source material it seems that the scores with salient segmentation using TokenCut omit comparisons with post processing using the bilateral solver proposed in the original paper. The reasoning behind this choice seems unclear, and we would appreciate an elaboration, given that the goal of upscaling is very much aligned with the authors proposed method.
>
> > TokenCut demands substantial memory resources when applied to a larger number of ViT tokens
>
> > Notably, this improvement is constrained by the model architecture, as TokenCut operates at the coarse segmentation level of ViT tokens. Given that SRT has the capability to provide finer-grained features (directly applying TokenCut at this level is computationally impractical due to its $\mathcal O(n^2)$ complexity, where $n$ is the number of tokens)
>
> Extracting the lowest two eigenvalues can be done efficiently with `torch.lobpcg`. This significantly reduces computational overhead, and we are uncertain as to the validity of the complexity estimate provided by the authors.
>
> We kindly ask that the authors address these concerns in their rebuttal.

---

> ### Author Response · Authors · 2023-11-16
> **Author response to the initial review**
>
> **W1:** *insufficiently motivated ... the exact nature of these artefacts*
>
> **R1:** We are not sure about the meaning of the term “exact nature”, but for imaging data the two most salient phenomena are occlusions and scale. Neither are respected by spatial quantization: occlusions are blended in samples that straddle the boundary, depriving the representation of precious information about the topology of the scene. Scale is also critically lost since the same spatial quantization of the image may correspond to any portion of an object, depending on its distance. This makes it difficult for the learned representation since it is forced to learn that the same object can span anywhere from a single sample to the entire image.
>
> In practice, the shortcomings of quantization schema are evident in DINO-based methods for segmentation, e.g. Tokencut, which only creates low-resolution segmentation due to quantization, and requires post-processing and refinement of pixel-level segmentation maps.
>
> This seems to be sufficient motivation to us, but if there are other aspects of spatial quantization that we missed we are certainly open to elaborate. We are more than happy to revise the paper to make the motivation clear to the readers.
>
>
> **W2:** *simpler interpolation methods exist... not clear if these new feature maps can be said to be faithful...*
>
> **R2:** We display the feature map only as a way of visualizing the outcome of SRT, which is a feature map that is strictly more informative than the original quantized map. It is difficult to determine whether feature maps are  “faithful” (and to what degree of “faithfulness”), so we measure it by the improvement on downstream tasks relative to the baseline. This is particularly the reason for conducting our experiments under zero-shot settings: Given that the encoding and decoding layers are frozen, features that are not faithful to the original would disrupt the outcome. On the contrary, our experiments show that the ensembled features yield consistent improvement for a number of downstream tasks including video object segmentation, depth prediction, unsupervised saliency segmentation, image retrieval and unsupervised object discovery; while it is difficult to define “faithfulness”, experimental results imply that the ensembled features are not “unfaithful”.
>
> Moreover, please refer to the Appendix of the revised manuscript for a comparison with (bilinear) interpolation. We compare SRT and simple interpolation to features from single forward passes. It is evident that SRT makes adjustments to the feature map at semantic boundaries, whereas simple interpolation displays a less meaningful grid pattern. We hope that this provides further support for the 'faithfulness' concern raised by the reviewer.
>
> **W3:** *conceptual overlap with Amir et al. (2021)...*
>
> **R3:** Indeed, Amir et al. is only conceptually related and direct comparisons cannot be made: We could compare Amir et al. to our result prior to aggregation, but that would defeat the purpose because our goal is to retain the same dimension of the original representation. Amir et al. requires making modifications to the forward pass by using a stride smaller than the quantization domain to produce overlapping tokens that ultimately yields 2x increase in spatial resolution (but with 4x more tokens); whereas, our method does not require any changes to the forward pass and produces 16x increase in signal resolution. To match the same granularity, Amir et al. would need to produce 256x more tokens (65536x more compute), which makes it computationally infeasible. Nevertheless, we reproduced the results of Amir et al. and tested their 2x super-resolved ViT-S/16 feature maps on video object segmentation (Table 1, main paper) following protocol specified in DINO. Their features resulted in worse performance than the baseline pretrained model, whereas with just 1 pixel of displacement, SRT improves over the baseline (Fig. 3, main paper).
>
> **W4:** *Methodology for non-dense downstream tasks unclear...*
>
> **R4**: Reproducibility will be facilitated by open-sourcing our code which we will do upon completion of the review process. In the meantime, We have added details about input resolutions as requested by the reviewer in our revised manuscript. For evaluation on retrieval, we follow protocols used in DINO and use the original resolution from the datasets.
>
> **W5:** *The link to stochastic resonance...*
>
> **R5:** This is a great suggestion. We will emphasize as suggested
>
> **W6:** *the study is limited to ViT-S16*
>
> **R6:** SRT is not limited to ViT-S/16 capacities. We do show results for ViT-B/16 in Table 1 of the main paper. Additionally, in Table 2, we show results on VIT-B/14. In fact, we have no restriction in network capacity as we model the operation of inverse translation and average pooling into a recursive mean. So, SRT increases negligible memory footprint; however, as with all ensembling methods, it does scale in time.

---

> ### Author Response · Authors · 2023-11-16
> **Author response to the initial review (continued)**
>
> **W7:** *While the method is novel, the current applications seem moderately niched.*
>
> **R7:** Respectfully, we disagree: In the main paper, we demonstrated five diverse applications of SRT: video object segmentation, depth prediction, unsupervised saliency segmentation, image retrieval, and unsupervised object discovery. We have additionally added classification and semantic segmentation in Table 5 and 6 of the Appendix, respectively, for a total of seven tasks covering a broad range of computer vision applications. SRT shows promise for all of them.
>
> We acknowledge the reviewer's perspective, as our emphasis on SRT's ability to achieve fine-grained feature maps may have been more pronounced. In response to this feedback, we plan to revise the paper to give greater emphasis to the practical applications of SRT.
>
> **Q1：** *What is the precise nature of the artefacts the authors wish to remedy?*
>
> **R8:** Spatial quantization causes information loss about key phenomenologies such as occlusion and scale, which we expect to impact performance in geometric and semantic inference tasks. Therefore, we expect that the artifacts of quantization will be manifest in all downstream visual tasks, and we test this assumption on different vision tasks.
>
> **Q2：** *Have the authors considered adding metrics for evaluating the faithfulness of the attributions in the visualisations of the features?*
>
> **R9:** Visualization is only for illustrative purposes, since the underlying representation is high dimensional. We use downstream metrics to evaluate the faithfulness of the representation to the content of the scene, despite quantization of the image. If there are other metrics thats evaluate the faithfulness we are also happy to include in the revised paper.
>
> **Q3：** *Why were contrastive comparisons to Amir et al. (2021) (in terms of results) omitted?*
>
> **R10:** (Also see R8) Amir's approach is conceptually related, but in reality, their original work increases feature size by 2 times (equivalent to 4 times the number of tokens). In contrast, our method achieves a 16 times denser feature map. Even when adhering to their original settings, we encounter hardware limitations on most tasks. Interestingly, in Table 1, simply increasing the number of patches to overlapping patches is detrimental. We hypothesize that this practice alters the forward pass of the Vision Transformer (ViT), generating features outside the domain for which the original ViT is trained. In contrast, SRT does not modify the forward pass.
>
> Method | F&J-Mean | J-Mean | J-Recall | F-Mean | F-Recall
> ----------- | ----------- | ----------- | ----------- | ----------- | ----------- |
> Baseline (DINO-ViT-S/16) | 0.617 | 0.602 | 0.740 | 0.634 | 0.764
> +Overlapping Tokens  (Amir et al.)  | 0.591 | 0.577 | 0.706 | 0.605 | 0.741 |
> SRT | **0.642** | **0.632** | **0.783** | **0.653** | **0.819** |
>
> **Q4：** *Exactly which embeddings were used for the KNN in the image retrieval task? The ensemble of class tokens? Pooled features? This is very unclear.*
>
> **R11** We follow the protocol of the original DINO paper, which applies retrieval on the class token. Note that in order to obtain ensemble class token with SRT, we do not naively average the class token of augmented image, but ensemble PRIOR to the attention mechanism in the last layer. In this way the final class token is computed from the ensemble SRT feature. We have revised the paper to add the details as requested by the reviewer.
>
>
> **Q5：** *The metrics for results in table 4, are they mAP? What does the columns 1-6 refer to?*
>
> **R12** Yes mAP. 1-6 refer to noise level in SRT. We have revised the paper to make it clear to the readers.
>
> **Update:** As suggested by the reviewer, we will include a discussion on *W3/R3* in the paper.

---

> ### Author Response · Authors · 2023-11-16
> **Author response to concerns about running TokenCut for saliency region segmentation**
>
> **Reviewer:** *"we are uncertain as to the validity of the complexity estimate provided by the authors."*
>
> **Our response regarding computational constraints:**
> We have confirmed the $O(n^2)$ complexity with the original authors of TokenCut through email. There might be a misunderstanding, but this complexity arises not from eigenvalue computation but due to TokenCut's use of graphcut techniques, necessitating the construction of a fully connected graph among the patches (implemented by Numpy on CPU). Hence, like other graphcut-based methods, the computational feasibility of TokenCut depends largely on the size and granularity of the features. In the case of ViT-B architectures, the feature dimensions increase from 256 to 768. The hardware constraint we faced upon paper submission was from the CPU and RAM, not the GPU. ViT-B/8 has 4x more tokens. To build a complete graph on ViT-B/8, one would create 16x more graph connections each requiring 3x more memory. To the best of our ability (and the limitations of our compute), we have tried to conduct experiments with TokenCut on our ViT-B/16 and ViT-B/8 features but were only able to run it on ViT-S/16 (Table 3 in the main text). We have clarified this point in the revised paper (Sec. 3.4).
>
> Upon rebuttal, with a hardware update, we managed to resolve the issue and successfully ran ViT-B/16 with TokenCut, the results are as follows and we are happy to include the results in the revision:
>
> Datasets | Method | maxF | IoU | Accuracy |
> ----------- | ----------- | ----------- | ----------- | ----------- |
> ECSSD | Baseline | 80.3 | 71.0 | 91.5 |
> ECSSD  | SRT | **81.8** | **72.6** | **92.2** |
> DUTS | Baseline | 66.4 | 56.7 | 89.5 |
> DUTS  | SRT | **68.8** | **58.3** | **90.6** |
> DUTS-OMRON | Baseline | 56.7 | 50.5 | 85.4 |
> DUTS-OMRON  | SRT | **58.0** | **51.6** | **86.1** |
>
> It is worth noting that, even with updated hardware, directly applying TokenCut to the fine-grained features is still computationally infeasible, since the number of tokens increases from $(h/16) \times (w/16)$ to $h \times w$ (256 times more), thus the computational resources required to handle the fully-connected graph becomes intractable. As stated in the paper, "we anticipate that future research will develop methods to leverage SRT’s high-resolution embeddings effectively."
>
> ___
> **Reviewer:** *"the scores with salient segmentation using TokenCut omit comparisons with post processing"*
>
> **Our response regarding postprocessing:**
> It's important to emphasize that our goal is to utilize TokenCut as a metric for comparing the original ViT feature with the ensemble SRT feature, rather than enhancing TokenCut itself. While post-processing techniques like Bilateral Solver or CRF refine segmentation boundaries, they don't directly address the comparison. As stated in the paper 'We execute TokenCut without any post-processing, such as Conditional Random Fields (CRF), to assess the raw quality of ViT embeddings.' (Sect. 3.4) Furthermore, our experiments are conducted fairly, using pooled features that share the same resolution as the original features employed by TokenCut.
> Nevertheless, as requested by the reviewer, below we provide comparison of TokenCut with Bilateral Solver and ViT-S/16:
>
> Datasets | Method | maxF | IoU | Accuracy |
> ----------- | ----------- | ----------- | ----------- | ----------- |
> ECSSD | Baseline | 87.4 | **77.2** | 93.4 |
> ECSSD  | SRT | **88.4** | 77.0 | **93.6** |
> DUTS | Baseline | 75.5 | **62.4** | 91.4 |
> DUTS  | SRT | **76.5** | **62.4** | **91.7** |
> DUTS-OMRON | Baseline | 69.7 | 61.8 | 89.7 |
> DUTS-OMRON  | SRT | **70.6** | **62.4** | **89.9** |
>
> We hope this answers the reviewer's question. If there are further concerns, we are more than willing to provide additional explanations.

---

> ### Comment · Reviewer_wDbQ · 2023-11-18
>
> > We are not sure about the meaning of the term “exact nature” [...]
>
> > Spatial quantization causes information loss [...]
>
> The exact nature of the term exact nature is, in this reviewers mind, semantically unambiguous. Specifically, this reviewer was referring to a formal exposition, preferably in more formal mathematical terms to heighten the clarity of the work. The discussion of the spatial quantisation and the link to stochastic resonance requires a clear exposition of the artefacts you discuss, with a level of formality. As you may have noticed, this reviewer is not in lack of understanding, but asking for specifications which could heighten the readability of the paper.
>
> *This may have been addressed in the revision*, however it seems that reviewers currently cannot see the revision history. This reviewer have voiced their concerns in the appropriate channels to see if if this is, for some reason, intentional, or a hiccup in the system.
>
> > It is difficult to determine whether feature maps are “faithful” (and to what degree of “faithfulness”) [...]
>
> > Visualization is only for illustrative purposes, [...]
>
> Several metrics for faithfulness of model attributions exist, e.g. *"A Comparative Study of Faithfulness Metrics for Model Interpretability Methods"* (Chan et al. 2022). As you are claim these attributions as a leading contribution ("yields a versatile visualization tool"), it would be diligent to address this, or simply adjust the significance of this precise contribution.
>
> > Amir et al. is only conceptually related and direct comparisons cannot be made [...]
>
> > Amir's approach is conceptually related, but in reality, their original work increases feature size [...]
>
> It seems that a overarching contrastive comparison can indeed be made by virtue of the discussion included by the authors. This discussion in your response is important for readers of your paper, and precisely the reason for our bringing light to the matter. Can the authors be compelled to expand their related work, or include the discussion in the appendix?
>
> > Reproducibility will be facilitated by open-sourcing our code [...]
>
> While your dedication to sharing your code is admirable, it is no replacement for detailing your methodology, since a practitioner starting from scratch to reproduce your results will potentially avoid the artefacts that could occur due to a specific implementation.
>
> > R5: This is a great suggestion. We will emphasize as suggested
>
> We are happy to have positively contributed to your work. Ideally this ties in with our concern on the nature of quantisation artefacts.
>
> > SRT is not limited to ViT-S/16 capacities
>
> Apologies if the comment seemed to imply that your method could not be applied for other capacities. Specifically, we meant to address base capacities in the segmentation results, which now has been dutifully addressed in a later comment.
>
> > Respectfully, we disagree: [...] (on niched applications)
>
> it is fine that the authors disagree, but other methods to handle quantisation artefacts exist, notably recent works on adaptable tokenisation, e.g. *"Vision Transformers with Mixed-Resolution Tokenization"* (Ronen et al. 2023) which tackles the same issue by changing the tokenisation method in-place. While this by no means diminishes the authors work, since the approaches are clearly different, we note that such methods have more direct applications. A discussion on alternative approaches to tackle the issues of uniform square spatial quantisation would serve to add context to the work and ongoing research in the field. Particularly since it is also can be applied in conjunction with SRT.
>
> > Note that in order to obtain ensemble class token with SRT, we do not naively average the class token of augmented image, but ensemble PRIOR to the attention mechanism in the last layer
>
> This ties in with our concerns about the methodology of the work in previous replies. While this could perhaps be made clear by surveying the code, this reviewer suggests that *it is imperative that the methodology is made explicit in the paper*. While the authors are naturally concerned with space constraints, a full exposition in the appendix is not only preferable, but necessary.
>
> **On a general note**: We note that the authors have made efforts to improve their work, and we are happy to see clarification in the presentation of results and inclusion of more contrastive experiments for their approach. Our remaining concerns are related to the methodology, reproducibility and context for readers to ascertain the overall value of the contribution.
>
> **Update**: To clarify our position, at this time this reviewer tends to keep their original rating.

---

> > ### Author Response · Authors · 2023-11-22
> > **Official Comment by Reviewer wDbQ: Added Appendix E for clarification**
> >
> > We apologize for misunderstanding the request for a more formal exposition of the origin of the quantization artifacts. We have now added an appendix (Appendix E) to describe how the artifacts originate, and how SRT reduces them, leveraging the averaging properties of SR. Hopefully the exposition gives more clarity to the motivation for SRT, but we caution that formalizing the process requires somewhat heavy notation since the artifacts arise from the interplay of the central projection, quantization, the piecewise smooth geometry of the underlying surfaces, and the piecewise smooth statistics of the underlying albedo or BRDF. We did not include this in the main paper since only readers who are familiar with computer vision would appreciate it and find it useful to develop better insight, so we leave it in the appendix in hope that it will help those with the expertise to better contextualize our contribution. As a summary, the most elementary image formation process that describes the phenomenology of interest uses piecewise smooth surfaces $S$ supporting piecewise smooth albedo $\rho$ (or BRDF $\beta$, akin to a computer graphics modeling pipeline of circa 1970s). Once projected through central perspective and quantized into a lattice (their composition is the operator $\pi$), the irradiance field $\tilde x$ is piecewise constant, with discontinuities at pixel boundaries $\Lambda$, entirely unrelated to the discontinuities of either the underlying shape $S$ or albedo $\rho$. A patch-based tokenizer aggregates a collection of neighboring pixel values into a vectorized representation $\phi(\tilde x)$, which in theory can retain all the information in the pixels in $D_i \ni \tilde x$, but it cannot retain information about the discontinuities in the scene, since the same patch, taken from a slightly different vantage point, subtends different portions of the scene. In other words, there is averaging (information loss) even if the dimension of the embedding is larger than the dimension of the patch. By sampling the patches at multiple locations $T_i$, we can recuperate (partial) structure of the underlying scene up to the level of granularity of the sampling $\min d(T_i, T_j)$, rather than granularity of the tokenization $|D_i|$. Even after averaging sampled-transformed patches to retain the same dimensionality of the patch embedding, we can clearly see better resolved structures of the scene such as occluding boundaries and albedo boundaries, not visible in the original patch-based representation, nor any super-resolved version of it based on interpolation or generic priors.

---

> > > ### Comment · Reviewer_wDbQ · 2023-11-23
> > > **Thank you for detailed response**
> > >
> > > The last revision of the manuscript is well formulated, and looks to formally link stochastic resonance operators to the method and overall presents the framework in a more coherent and cohesive manner. This addresses a central concern for this reviewer, and raising the recommendation to a 6 as a result. Pardon the brevity of the comment due to time constraints.

---

> ### Comment · Reviewer_wDbQ · 2023-11-18
> **Brief comment on computational constraints**
>
> > Our response regarding computational constraints [...]
>
> We are glad to see our constructive feedback prompted extra results for your work.
>
> As a side note, there is no compelling reason for using NumPy to apply TokenCut. The full graph is essentially constructed by taking cosine similarity, thresholding it, and computing the Laplacian $L = D - A$. The classic normalised graph cut is essentially performed using the Fiedler vector (the eigenvector corresponding to the second lowest magnitude eigenvalue). All this is possible to apply on GPU without issue (PyTorch has all the tools necessary), and while the theoretical complexity might be high, it is on the same level as self-attention operators (less so since it does not require Q,K,V projections).

---

### Author Response · Authors · 2023-11-15
**A general message to the reviewers and AC**

We appreciate the reviewers for their valuable feedback. We are encouraged by the reviewers' positive response to the novelty of the paper. In response to the initial reviews, we have uploaded a revised paper. Here, we would like to address concerns about super-resolution and elaborate on how SRT obtains fine-grained feature maps through *ensemble* and how it differs from the conventional approach of *super-resolution*.
___
**Difference between super-resolution and ensembling:**

Given a signal $x$ that is sub-sampled to $\tilde x$, super-resolution aims to retrieve an approximation $\hat x$ of $x$ given $\tilde x$. Since information is lost in the sampling, the reconstruction depends crucially on the choice of prior (model, kernel, regularizer, etc.) which is arbitrary. Super-resolution is a form of hallucination: Attribute details to $\hat x$ that are not in $\tilde x$, in the hope that they will somehow match those in $x$. This requires strong faith in prior knowledge about $x$, $P(x)$.

Given the same signal $x$, one could instead generate multiple samples $\tilde x_i$, each with a different kernel, and then reconstruct a single estimate from the samples $\hat x = F(\tilde x_1, \dots, \tilde x_N)$. In our case, the kernel is fixed, so one can use any employed in super-resolution techniques, but it is transformed by a group action (translation). More general transformations (e.g. rotation, affine) could do as well. Now, the estimator $F$ has more information available about $x$ than in super-resolution: The sigma-algebra spanned by the random variables $\tilde x_i$ is a superset of the (trivial) sigma algebra spanned by the single sample $\tilde x$ in super-resolution.

In other words, SRT which aggregates different samples from a process contains more information than any single sample about the process.

___
**Revisions to the manuscript:**

We have made updates to the manuscript based on the reviewers' feedback and uploaded it. The key revisions include:

1. We incorporated results using overlapping patches, reproducing Amir et al., as suggested by Reviewer wDbQ. These results have been added to Table 1.
2. Following Reviewer wDbQ's suggestion, we included results with the bilateral solver as post-processing for unsupervised saliency segmentation, and these results are now included in Table 3.
3. We provided additional details on dataset resolution, and revised Table 4, as recommended by Reviewer wDbQ.
4. In response to Reviewer j1cP's suggestion, we added a comparison and discussion on feature resizing in Sec. C, Fig. 4, and Table 7 in the appendix.
5. As per Reviewer 3VWw's recommendation, we included results on ResNet with classification and semantic segmentation in Table 5 and 6 in the Appendix.
6. We added visual examples in Fig. 4 illustrating the differences between features from SRT and noisy features from a single ViT pass.

**Update:**

7. We added a pseudo-code of SRT, in response to Reviewer wDbQ.
8. We added additional formalization in Appendix E, in response to Reviewer wDbQ.
9. Additional visualization of ViT features, in response to Reviewer 3VWw.


Moving forward, we plan to revise the introduction, particularly focusing on motivation, the connection to stochastic resonance, and technical insights. These revisions will be made after the author-reviewer discussion period is finalized, since adjustments may be made based on the reviewers' further feedback. We are open to discussions with the reviewers if there are additional concerns.

---

### Meta-Review · Area_Chair_K1aw · 2023-12-02

**Metareview:**

**Summary:** The paper presents the Stochastic Resonance Transformer (SRT), a method enabling finer detail capture in Vision Transformers (ViTs). SRT applies perturbations to input images to avoid detail loss during the tokenisation process. This approach improves sub-token resolution in feature extraction, showcased across several computer vision tasks. However, the method's clarity, reproducibility, and applicability could be further explored according to the reviewers.

During the rebuttal phase, reviewers analyzed both the strengths and weaknesses of the paper, reaching a consensus that it is a borderline paper. The paper presents interesting ideas for the community, but it is not without significant issues that need to be addressed. Although initially receiving a high score from one reviewer, their confidence dwindeled through further discussion. Ultimately, the weaknesses appear to overshadow the strengths. A more meticulous revision by the authors would greatly benefit the paper. Therefore, I cannot advocate for its acceptance in its current form.

**Strengths:** The paper introduces a simple, versatile method for improving Vision Transformers by super-resolving feature embeddings, beneficial for dense prediction tasks. This post-hoc technique is architecture-agnostic, implying broad applicability for enhancement and fine-tuning across various vision tasks. The method, which extends to potential use in self-supervision, addresses a common limitation in ViTs related to uniform partitioning. It has been effectively demonstrated in multiple tasks, suggesting its robustness and potential for wide-ranging applications.

**Weaknesses:**  The paper received numerous criticisms from reviewers, particularly about its methodology, which appears to be under-explained and not fully reproducible. Furthermore, the visualizations provided are seen as somewhat questionable and need more quantitative measures. The effectiveness of the perturbations used weren't adequately detailed or convincing, making the overall utility of the translation less persuasive. Another criticism was the unclear articulation of the nature of artifacts, which could have been clarified better. While the authors provided a detailed formalization in Appendix E to address this, its complexity may have been excessive. Overall, several points of critique remain unresolved, casting doubt on the methodology, its explanation, and the paper's reproducibility.

**Justification For Why Not Higher Score:**

The main reasons for rejecting the paper center around unclear methodology, inadequate validation of visualizations, unresolved issues with regard to perturbations, and insufficient explanation of identified artifacts. Notably, the authors fail to provide crucial details for their methodology, including the number of forward passes required for their method. The visualizations, while an integral part of the paper's appeal, lack necessary quantitative validation. Additionally, the authors offer limited information about the effectiveness of perturbations  prove unconvincing. The appendix does address artifacts but the over-complicated explanation and initial lack of clarity leaves concerns unalleviated. The reviewer discussion emphasized these collective issues, significantly undermining confidence in the paper's rigor and potential impact, subsequently leading to an overall recommendation for rejection.

**Justification For Why Not Lower Score:**

N/A

---

### Decision · Program_Chairs · 2024-01-16

Reject